# Knapsack RL: Compute-Efficient Reinforcement Learning via Heterogeneous Rollout Allocation

**Ziniu Li** [1] [2]  **Congliang Chen** [3]  **Tianyun Yang** [4]  **Tian Ding** [4]  **Ruoyu Sun** [1] [4]  **Ge Zhang** [2]  **Wenhao Huang** [2]
**Zhi-Quan Luo** [1] [4]

## Abstract

Reinforcement learning (RL) fine-tuning of Large Language Models (LLMs) is compute-intensive because each prompt requires generating multiple responses, or rollouts. To make the best use of GPU resources, the key question is **how to allocate rollout-generation jobs across prompts**. Existing methods typically use *uniform* allocation, assigning every prompt the same rollout budget. This is inefficient and ineffective: easy prompts are oversampled after they are already solved, while hard prompts receive too little exploration. In policy optimization methods such as Group Relative Policy Optimization (GRPO), both cases often yield near-zero gradients and limited learning progress. We address this problem by formulating rollout allocation as a compute-constrained resource allocation task, where each prompt-budget choice has an expected learning value and rollout cost. Based on this view, we propose *Knapsack RL*, a practical framework that uses knapsack optimization to assign *heterogeneous*, task-specific rollout budgets. It directs exploration toward prompts that benefit most from additional sampling. Applied to GRPO, Knapsack RL increases the effective-gradient ratio by up to 40%, enables larger budgets for challenging prompts, and improves mathematical reasoning by 2–4 points on average, with peak gains of up to 9 points. Achieving similar performance with uniform allocation requires about $2\times$ more compute, demonstrating a simple and practical path toward scaling RL fine-tuning for LLMs.

---

[1]The Chinese University of Hong Kong, Shenzhen [2]Bytedance Seed [3]Shenzhen Loop Area Institute [4]Shenzhen International Center for Industrial and Applied Mathematics, Shenzhen Research Institute of Big Data. Correspondence to: Ziniu Li <ziniuli@link.cuhk.edu.cn>, Ge Zhang <zhangge.eli@bytedance.com>.

*Proceedings of the $43^{rd}$ International Conference on Machine Learning*, Seoul, South Korea. PMLR 306, 2026. Copyright 2026 by the author(s).

## 1. Introduction

Reinforcement learning (RL) has become a central paradigm for improving Large Language Models (LLMs) (Ouyang et al., 2022; Li et al., 2024; Shao et al., 2024; Guo et al., 2025). A particularly successful setting is RL with verifiable rewards, where models generate candidate solutions and receive outcome-based feedback from automatic verifiers (Lambert et al., 2024; Li et al., 2025c). This framework has driven rapid progress in mathematical reasoning (Jaech et al., 2024; Shao et al., 2024; Guo et al., 2025; Yu et al., 2025), and has also been extended to coding and agentic tasks (Luo et al., 2025a; Kimi Team et al., 2025). Unlike supervised fine-tuning, RL repeatedly invokes the current policy to generate multiple responses, or rollouts, for each prompt. In practice, rollout generation consumes a substantial share of GPU compute resources in large-scale LLM RL (Fu et al., 2025; Qin et al., 2025).

Improving RL compute efficiency has therefore become increasingly important. Recent work on RL infrastructure and asynchronous training improves *systems efficiency* by reducing pipeline bubbles, overlapping generation and training, and increasing hardware utilization (Fu et al., 2025; Lu et al., 2025; Sheng et al., 2025a). However, high GPU utilization does not necessarily imply high learning efficiency: even when GPUs are busy, the generated rollouts may provide little useful optimization signal. Thus, beyond systems throughput, there remains an *algorithmic learning efficiency* question: how can a fixed amount of rollout compute be converted into as much learning progress as possible?

This issue is especially visible in Group Relative Policy Optimization (GRPO) (Shao et al., 2024). GRPO obtains informative prompt-level gradients only when a rollout group contains both successful and failed responses. If all responses are correct, the prompt is already saturated; if all are incorrect, the model receives no contrastive signal (Yu et al., 2025; Chen et al., 2025a). In both cases, the rollout group yields a near-zero gradient, wasting the corresponding compute. Existing work addresses this issue through dynamic sampling, difficulty-aware filtering, prompt curriculum learning, and related mechanisms (Yu et al., 2025; Sun et al., 2025b; Qu et al., 2025; Gao et al., 2026; Shen

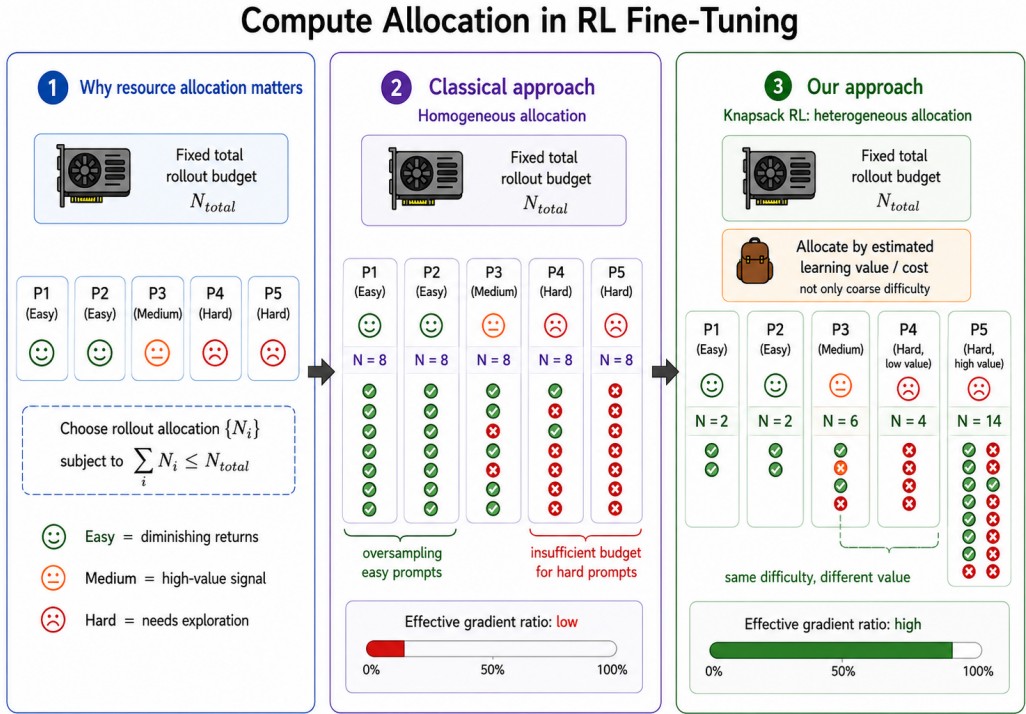

Figure 1. Overview of Knapsack RL. Given a batch of prompts and a fixed global rollout budget, the method estimates prompt-level learning value, solves a knapsack-style allocation problem, and generates heterogeneous rollout groups for policy optimization.

et al., 2026; Mao et al., 2026). These methods mainly decide *which prompts* should be selected, filtered, or prioritized.

In this paper, we study a complementary and largely under-explored axis: **how many rollouts should each prompt receive under a fixed global budget?** Most RL pipelines use *homogeneous allocation*, e.g., $N = 8$ rollouts for every prompt, regardless of difficulty. This rule is simple and system-friendly, but can be inefficient. As shown in Section 3, medium-difficulty prompts may yield mixed outcomes with only a few rollouts, while hard prompts may require much more exploration before any successful trajectory appears. Thus, the same fixed budget can be excessive for some prompts and insufficient for others.

This perspective motivates a broader view of rollout generation as a compute-constrained resource allocation problem (Hurwicz, 1973; Katoh et al., 2013). Each prompt-budget choice has a *cost*, corresponding to GPU rollout compute, and a *learning value*, corresponding to the expected usefulness of the resulting training signal. Under this view, uniform allocation is only one possible policy, and not necessarily an efficient one. A better policy can reassign rollout compute from low-value prompt-budget pairs to high-value ones, improving learning progress without increasing the global rollout budget.

Building on this view, we propose **Knapsack RL**, whose

workflow is illustrated in Figure 1. For each prompt, assigning a particular number of rollouts is treated as an item with a cost and a learning value, and a knapsack-style optimization maximizes total value under the same global rollout budget used by homogeneous allocation (Mathews, 1896; Pisinger & Toth, 1998). Only the allocation of rollout jobs changes; inference, reward evaluation, and policy optimization remain largely unchanged.

Empirically, we instantiate this framework for GRPO and evaluate it on mathematical reasoning benchmarks using Qwen-series and DeepSeek-R1-Distill models (Yang et al., 2024; 2025; Guo et al., 2025). Knapsack-GRPO consistently improves over homogeneous GRPO under the same total rollout budget. It increases the effective-gradient ratio by up to 40%, assigns larger rollout budgets to prompts that benefit from additional exploration, and improves reasoning performance by 2–4 points on average, with peak gains of up to 9 points on individual benchmarks. Moreover, our compute-scaling experiments show that achieving comparable performance with homogeneous allocation requires roughly $2\times$ more rollout compute. These results suggest that heterogeneous rollout allocation is a practical path toward compute-efficient RL fine-tuning for LLMs. Code: https://github.com/liziniu/KnapsackRL.

## 2. Preliminary

We introduce the RL fine-tuning setup and the GRPO estimator (Shao et al., 2024) used throughout the paper. Let $x$ denote an input prompt and $y \sim \pi_\theta(\cdot \mid x)$ denote a response sampled from the current policy. We focus on RL with verifiable rewards, a common setting for mathematical reasoning and other automatically checkable tasks (Lambert et al., 2024; Guo et al., 2025; Yu et al., 2025). Let $y = (\texttt{CoT}, \texttt{answer})$ denote a generated response with Chain-of-Thought reasoning (Wei et al., 2022) and a final answer. The reward is given by an automatic verifier:

$$r(x, y) = \mathbb{I}(\texttt{answer is correct with respect to } x), \quad (1)$$

where $r(x, y) \in \{0, 1\}$. Such binary outcome rewards are simple and effective when ground-truth answers or automatic checkers are available (Shao et al., 2024; Guo et al., 2025; Wen et al., 2025).

**Rollout groups.** For a mini-batch of $M$ prompts $\{x_1, \ldots, x_M\}$, let $N_i$ denote the number of responses sampled for prompt $x_i$. The sampled responses are

$$\{y_{i,1}, \ldots, y_{i,N_i}\}, \quad y_{i,j} \sim \pi_\theta(\cdot \mid x_i).$$

Most RL fine-tuning pipelines use a homogeneous group size, setting $N_i = N$ for all prompts. This paper studies the more general case where $N_i$ can vary across prompts.

**GRPO with variable group sizes.** We instantiate our framework with GRPO, which computes relative advantages within each prompt group. To support heterogeneous rollout allocation, we write the GRPO estimator with variable group sizes $\{N_i\}_{i=1}^M$:

$$g(\theta) = \sum_{i=1}^{M} \frac{1}{\sigma_i} \sum_{j=1}^{N_i} \nabla_\theta \log \pi_\theta(y_{i,j} \mid x_i) \cdot \left(r(x_i, y_{i,j}) - b_i\right),$$

$$(2)$$

where

$$b_i = \frac{1}{N_i} \sum_{j=1}^{N_i} r(x_i, y_{i,j}), \ \sigma_i = \sqrt{\frac{1}{N_i} \sum_{j=1}^{N_i} \left(r(x_i, y_{i,j}) - b_i\right)^2}.$$

Here $b_i$ is the prompt-level reward baseline and $\sigma_i$ is the within-group reward standard deviation, both computed over the $N_i$ rollouts assigned to prompt $x_i$. In implementation, we evaluate the normalization factor as $1/(\sigma_i + \epsilon)$ with a small numerical stabilizer $\epsilon$. If $\sigma_i = 0$, all centered rewards are already zero, so this stabilizer does not create an artificial training signal and the prompt contributes zero gradient. When $N_i$ varies, the gradient estimator remains unbiased.

## 3. Diagnosing Compute Inefficiency

We now diagnose why homogeneous rollout allocation can be inefficient from the perspective of algorithmic learning efficiency. Modern RL systems can improve hardware uti-

lization by overlapping rollout generation and model updating (Fu et al., 2025; Lu et al., 2025; Sheng et al., 2025a). However, keeping GPUs busy does not guarantee that the generated rollouts produce useful optimization signal. This section is deliberately diagnostic: we first identify rollout groups with *zero* training value, giving a simple criterion for definite compute waste. For rollout data that can produce non-zero signal, its relative learning value is modeled later by the value-cost allocation objective in Section 4.

### 3.1. Motivation: When Rollout Compute Produces No Learning Signal

Each rollout job occupies GPU compute and memory, and its generated data is useful only if it contributes a training signal. GRPO gives a simple diagnostic for wasted rollout data: under binary rewards, a prompt group is informative only when it contains both successes and failures. If all responses receive the same reward, the relative advantage vanishes.

**Observation 1** (Zero-gradient rollout groups)**.** *Fix a prompt $x_i$ and suppose it is assigned $N_i$ rollouts. Let $g_i$ denote its prompt-level contribution to Equation* (2)*. If all $N_i$ sampled responses for $x_i$ receive identical rewards, then $r(x_i, y_{i,j}) - b_i = 0$ for all $j$, and hence $g_i = 0$.*

Thus, zero-gradient groups give a concrete compute-waste criterion: the system spends GPU memory, decoding compute, and reward-evaluation cost to produce rollout data, but the optimizer receives no prompt-level learning signal. This sparse-gradient issue has also motivated dynamic sampling and difficulty-aware filtering in recent LLM RL work (Yu et al., 2025; Sun et al., 2025b; Qu et al., 2025; Gao et al., 2026).

To quantify how much rollout compute is converted into optimizer signal, we define the metric of `effective-gradient-ratio`:

$$\text{EGR} := \frac{1}{\sum_{i=1}^M N_i} \sum_{i=1}^{M} \sum_{j=1}^{N_i} \mathbb{I}\left(g_{i,j} \neq 0\right), \qquad (3)$$

where $g_{i,j}$ denotes the gradient contribution of response $y_{i,j}$. A low EGR means that GPU resources are spent to generate rollout data that is not converted into useful gradient updates.

**Empirical observation.** We track this conversion for `Qwen2.5-Math-7B` trained on `DAPO-MATH-17K` in Figure 2. Each mini-batch contains $M = 256$ prompts and uses a homogeneous budget of $N = 8$ rollouts per prompt. We also separate two zero-gradient cases:

- `all-positive`: all rollouts are correct, so additional rollout jobs mostly oversample an already solved prompt;

- `all-negative`: all rollouts are incorrect, so the as-

signed budget fails to produce useful successful data.

These two cases correspond to different forms of compute waste: oversampling solved prompts and underexploring hard prompts.

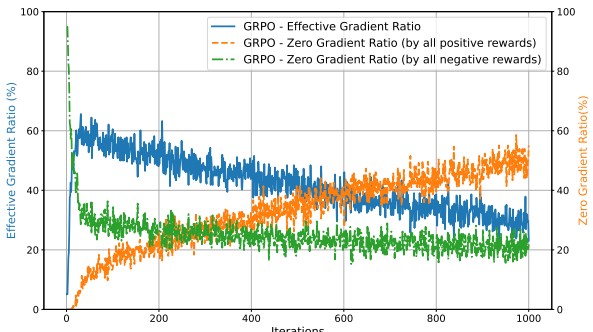

*Figure 2.* Effective-gradient ratio and zero-gradient prompt groups during homogeneous GRPO training.

The effective-gradient ratio remains below 60% for most of training, meaning that many rollout jobs produce data with zero optimizer signal. The dominant source of waste changes over time:

- **Early training.** Most wasted rollout jobs are all-negative: the budget is too small to discover successful trajectories.

- **Middle training.** More prompts enter the mixed-reward regime, where the same rollout budget is more likely to produce useful gradients.

- **Late training.** Waste becomes bimodal: easy prompts are oversampled as all-positive groups, while hard prompts remain underexplored as all-negative groups.

These dynamics reveal a mismatch between prompt difficulty and homogeneous rollout budgets. The same number of rollout jobs can be excessive for solved prompts but insufficient for prompts that need more attempts to yield useful data.

### 3.2. Theoretical Analysis: Difficulty Determines Rollout Demand

We next use this criterion to analyze how many rollouts are needed merely to make a prompt capable of producing a non-zero GRPO signal. This is a necessary signal condition, not a recommendation to assign large budgets to every easy or hard prompt. For a prompt $x_i$, define its current success rate as

$$p_i := \Pr_{y \sim \pi_\theta(\cdot|x_i)} \big[ r(x_i, y) = 1 \big].$$

Assuming independent rollouts, the probability that a group of $N_i$ responses contains both successes and failures is

$$\Pr(g_i \neq 0) = 1 - p_i^{N_i} - (1 - p_i)^{N_i}. \tag{4}$$

The two subtracted terms correspond to the all-positive and all-negative events, respectively. This expression shows that the rollout budget required for a non-zero GRPO signal depends sharply on the prompt difficulty. When $p_i \approx 0.5$, a small number of rollouts is usually sufficient to observe both outcomes. When $p_i \approx 0$ or $p_i \approx 1$, many more rollouts may be required to avoid an all-negative or all-positive group.

**Proposition 1** (Rollout budget for non-zero gradients)**.** *Given a prompt with success rate $p \in (0, 1)$:*

- ***High-probability bound.*** *For any $\alpha \in (0, 1)$, a sufficient condition for $\Pr(g_i \neq 0) \geq \alpha$ is*

$$N_i \geq \frac{\ln\big((1 - \alpha)/2\big)}{\ln(\max\{p, 1 - p\})}.$$

- ***Expected rollouts until the first non-zero group.*** *If $N_i$ is the number of rollouts required until both outcomes are observed for the first time, then*

$$\mathbb{E}[N_i] = \frac{1}{p} + \frac{1}{1 - p} - 1.$$

Appendix B.1 gives the proof. We validate this prediction empirically in Figure 3. We use `Qwen2.5-Math-7B-Instruct` to generate 256 responses for 1,000 prompts from `DAPO-Math-17K`. We estimate each prompt's success rate $p_i$ and compute the rollout budget required to obtain non-zero signal. A standard budget such as $N = 8$ mainly covers prompts with intermediate success rates. Near $p_i = 0$ or $p_i = 1$, many rollouts may be needed to force a mixed-reward group; this does not imply that such prompts should always receive large budgets, since their marginal learning value is assessed later by the value-cost allocation objective.

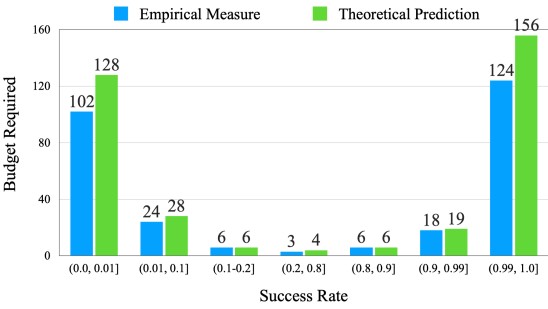

*Figure 3.* Rollout budget required to obtain non-zero GRPO gradients as a function of prompt success rate. Theoretical curves correspond to expected rollouts. Empirical bins aggregate prompts, so the plot need not be perfectly symmetric in $p$ and $1 - p$.

This analysis exposes the core compute-allocation dilemma. Increasing $N$ uniformly can help difficult prompts, but it also increases the rollout cost for every prompt, including prompts that are already easy or already informative with a small budget. Therefore, *homogeneous scaling can improve learning signal only by paying a large global compute cost.*

Although our experiment focuses on GRPO, the study is not limited to it. Other policy-gradient estimators, including REINFORCE (Williams, 1992), ReMax (Li et al., 2024), and RLOO (Ahmadian et al., 2024), also rely on sampled rollouts to obtain learning signals.

### 3.3. From Prompt Selection to Rollout Allocation

The diagnosis above suggests two complementary ways to improve the compute efficiency of RL fine-tuning: selecting more useful prompts, and allocating more appropriate rollout budgets to the selected prompts.

**Prompt selection.** The first approach changes *which prompts* are used for training. Dynamic sampling, difficulty-aware filtering, curriculum learning, and online difficulty prediction follow this direction (Yu et al., 2025; Sun et al., 2025b; Qu et al., 2025; Gao et al., 2026; Shen et al., 2026; Mao et al., 2026). These methods aim to avoid spending rollout compute on prompts that are predicted to be uninformative under the current policy, such as prompts that are already too easy or currently too hard to yield effective gradients. In this sense, prompt selection acts as an admission-control mechanism: it decides which prompts enter the rollout batch.

However, prompt selection alone can be restrictive when each selected prompt still receives a *fixed rollout budget*. A hard prompt may be rejected simply because a small fixed budget, such as $N = 8$, is unlikely to discover a successful trajectory, even though the prompt could provide high learning value if given more attempts. Conversely, an easy prompt may be kept in the batch but receive more rollouts than necessary. Thus, prompt selection addresses whether a prompt should be trained on, but not how much rollout compute it should receive.

**Rollout allocation.** The second approach changes *how many rollouts* each prompt receives after prompts have entered the batch (Yao et al., 2025). This is the problem studied in this paper. Rollout allocation is a graded resource-allocation decision rather than a binary selection decision. A hard prompt need not be discarded merely because a default budget is insufficient; it can instead receive a larger budget if its expected learning value justifies the additional cost. Similarly, an easy prompt need not be removed entirely; it may receive only a small budget for coverage.

These two approaches are conceptually distinct and practically compatible. Prompt selection controls the composition of the training batch, while rollout allocation controls the distribution of GPU decoding capacity within that batch. Prior selection methods can provide useful difficulty estimates or task-value signals for allocation, and our allocation method can be applied after a prompt subset has been selected. We therefore view prompt selection and rollout allocation as orthogonal design choices that can be combined. In this paper, we focus on the latter axis: given a fixed global rollout budget, how should rollout compute be distributed across prompts to maximize learning value?

## 4. Knapsack RL: Resource-Aware Rollout Allocation

We now present **Knapsack RL**, a resource-aware framework for assigning heterogeneous rollout budgets across prompts. The diagnosis in Section 3 shows that homogeneous allocation can waste compute in two ways: easy prompts may be oversampled after they already produce reliable successes, while hard prompts may be undersampled and fail to produce any successful trajectory. This suggests that rollout generation should be treated not only as an exploration procedure, but also as a *resource management problem* (Hurwicz, 1973; Katoh et al., 2013).

### 4.1. Rollout Allocation as Resource Management

In each RL iteration, the trainer has a fixed amount of generation compute, denoted by $N_{\text{total}}$. Allocating $N_i$ rollouts to prompt $x_i$ consumes rollout compute and produces an expected learning benefit. We therefore associate each prompt-budget decision with a **cost** $C_i(N_i)$ and a **value** $V_i(N_i)$. In our main formulation, each rollout is treated as one unit of compute, so $C_i(N_i) = N_i$; this approximation can be replaced by token-level or latency-level costs in system-aware variants. The value $V_i(N_i)$ measures the expected learning utility of assigning $N_i$ rollouts to prompt $x_i$. The diagnostic in Section 3.1 only identifies zero-value rollout groups; here we further score non-zero-gradient groups by how useful their signal is for improving the policy.

Under this view, homogeneous allocation $N_i = N$ is only one possible policy. It is simple, but it implicitly assumes that all prompts have the same value-cost profile. Our goal is to reallocate the same total compute toward prompt-budget choices with higher expected learning value.

### 4.2. Knapsack Formulation

We formalize rollout allocation as a knapsack-style optimization problem. Given a mini-batch of $M$ prompts, we choose integer rollout budgets $N_1, \ldots, N_M$ by solving

$$\max_{N_1,\ldots,N_M} \quad \sum_{i=1}^{M} V_i(N_i) \tag{5}$$

$$\text{s.t.} \quad \sum_{i=1}^{M} C_i(N_i) \leq N_{\text{total}},$$

$$N_{\text{low}} \leq N_i \leq N_{\text{up}}, \quad N_i \in \mathbb{Z}^+.$$

Here $N_{\text{low}}$ enforces minimal coverage for each prompt, while $N_{\text{up}}$ prevents degenerate allocations that spend too much compute on a single prompt. In our implementation, $C_i(N_i) = N_i$, and $N_{\text{total}}$ is set to match the compute used by homogeneous allocation, i.e., $N_{\text{total}} = MN$.

This formulation is a discrete resource-allocation problem with a knapsack constraint (Pisinger & Toth, 1998). Each feasible prompt-budget pair can be viewed as an item: its cost is the rollout compute it consumes, and its value is the expected learning benefit it provides. The optimization selects one budget for each prompt while keeping the total rollout-count budget within the global budget.

### 4.3. Task Value for GRPO

We next define the value function used in Equation (5). For GRPO, a rollout group is useful only when it contains both successful and failed responses. Let

$$p_i := \Pr_{y \sim \pi_\theta(\cdot|x_i)} \big[r(x_i, y) = 1\big]$$

denote the current success rate of prompt $x_i$. Under independent rollouts, the probability that $N_i$ samples produce a non-zero GRPO gradient is

$$P_{\text{nz}}(N_i, p_i) = 1 - p_i^{N_i} - (1 - p_i)^{N_i}. \qquad (6)$$

This term captures the *gradient availability* induced by additional sampling.

However, a non-zero gradient is not equally valuable for every prompt. A prompt that is already easy may produce non-zero gradients, but additional updates on it may offer limited room for improvement. We therefore multiply the non-zero-gradient probability by an *information gain* term:

$$V_i(N_i) = P_{\text{nz}}(N_i, p_i) \cdot \text{InfoGain}(p_i). \qquad (7)$$

The first factor asks whether the assigned rollouts are likely to produce an effective gradient; the second factor asks whether such a gradient is expected to be useful.

We define $\text{InfoGain}(p_i)$ as the expected increase in success probability after one update. Let $p_i^t$ and $p_i^{t+1}$ denote the success rates before and after the update. Ideally,

$$\text{InfoGain}(p_i^t) = p_i^{t+1} - p_i^t. \qquad (8)$$

This ideal value would require generating the candidate rollouts, updating the policy, and evaluating the updated policy for each prompt-budget choice, which is computationally prohibitive inside every RL iteration. It also need not satisfy diminishing returns, making efficient greedy allocation difficult. We therefore use a first-order approximation that is cheap to compute and preserves a useful marginal-value structure. Since $p_i^{t+1}$ is unavailable before training, we approximate this quantity using the current success rate.

**Proposition 2.** *Under the Taylor approximation in Ap-*

*pendix B.2, the information gain can be approximated as*

$$\text{InfoGain}(p_i^t) \approx p_i^t (1 - p_i^t)^2.$$

Dropping the iteration superscript for clarity, the resulting GRPO value function is

$$V(N, p) := \big(1 - p^N - (1 - p)^N\big)p(1 - p)^2. \qquad (9)$$

This value function has two desirable effects. The term $1 - p^N - (1-p)^N$ favors budgets that make mixed outcomes likely, while the term $p(1 - p)^2$ gives higher learning value to challenging but still solvable prompts. In particular, the value is asymmetric between $p$ and $1 - p$: a hard prompt with success rate $p < 0.5$ receives larger value than its easy mirror prompt with success rate $1 - p$.

### 4.4. Marginal-Value Allocation

We next characterize the optimal allocation induced by the value function, which provides additional insight into how Knapsack RL distributes rollout compute. The key property of Equation (9) is that the value of additional sampling has diminishing returns. Define the marginal value of assigning one more rollout as

$$\Delta V(N, p) := V(N + 1, p) - V(N, p). \qquad (10)$$

For our value function, this marginal value is

$$\Delta V(N, p) = \big[p^N(1 - p) + (1 - p)^N p\big]p(1 - p)^2. \qquad (11)$$

**Proposition 3.** *For any $p \in (0, 1)$, the value function in Equation (9) satisfies:*

- **Monotonicity.** $V(N, p)$ *is non-decreasing in $N$.*

- **Diminishing returns.** $\Delta V(N, p)$ *is strictly positive and strictly decreasing in $N$.*

- **Hard-task bias.** *For any $N \geq 1$ and $p \in (0, 0.5)$,*

$$V(N, p) > V(N, 1 - p), \quad \Delta V(N, p) > \Delta V(N, 1 - p).$$

Appendix B.3 gives the proof. These properties imply that the optimal assignment can be understood through *marginal value*. Starting from the minimum allocation $N_i = N_{\text{low}}$, each additional rollout should be assigned to the prompt that benefits most from one more sample:

$$j = \arg\max_i \Delta V(N_i, p_i). \qquad (12)$$

The selected prompt receives one additional rollout, its marginal value is updated, and the process repeats until the total budget is exhausted.

**Proposition 4** (Greedy optimality; informal)**.** *For the unit-cost case $C_i(N_i) = N_i$, the optimal solution to Equation (5) can be obtained by greedily assigning each remaining rollout to the prompt with the largest current marginal value $\Delta V(N_i, \widehat{p}_i)$. A formal exchange argument is provided in Appendix B.4.*

This result highlights a key point: Knapsack RL does *not*

simply allocate more rollouts to every hard prompt, nor does it only target prompts near $p = 1/3$. Instead, it repeatedly asks which prompt has the largest *marginal learning value* from one additional rollout:

- **Small budgets.** The method often prioritizes prompts whose current success rates make mixed outcomes likely.

- **Larger budgets.** These prompts become saturated as their marginal gains decrease, so allocation shifts toward harder prompts where additional sampling can reveal rare successful trajectories.

Thus, the priority is dynamic: it depends jointly on prompt difficulty, current assigned budget, and remaining compute.

### 4.5. Practical Implementation

Implementing Knapsack RL requires estimates of the prompt success rates $p_i$. If such estimates are available during dataset construction, we directly use them in Equation (5). Otherwise, we estimate them online. In our experiments, the first epoch uses homogeneous allocation to collect initial reward statistics. For later epochs, we use the success rates estimated from the previous epoch as $\hat{p}_i$, solve the knapsack allocation problem, and then generate heterogeneous rollout groups according to the resulting budgets. We update these estimates at the epoch level.

The greedy solver is lightweight. With unit rollout costs, it can be implemented using a priority queue over the marginal values in Equation (11); in our experiments, the allocation step takes less than one second. The rest of the RL pipeline remains unchanged: rollout generation, reward evaluation, and policy optimization are performed as in standard GRPO. From a systems perspective, Knapsack RL only changes the number of rollout requests assigned to each prompt, and is compatible with existing inference and training components such as vLLM (Kwon et al., 2023), FSDP (Zhao et al., 2023), and Megatron (Shoeybi et al., 2019).

## 5. Experiments

### 5.1. Main Results

**Experiment Setting.** We implement Knapsack RL and baseline methods using the large-scale RL training framework `Verl` (Sheng et al., 2025b). Our primary focus is GRPO (Shao et al., 2024), a widely examined method, and we refer to our specific implementation as *Knapsack-GRPO*. Training utilizes the DAPO-Math-17K dataset (Yu et al., 2025), which comprises 17,917 prompts, each with a ground truth answer for verification. We conduct experiments with both pre-trained and instruction-tuned models. The pre-trained models include Qwen3-4B-Base (Yang et al., 2025) and Qwen2.5-Math-7B (Yang et al., 2024). For instruction-tuned models, we utilize DeepSeek-R1-Distill-Qwen-1.5B

(Guo et al., 2025) (abbreviated as DPSK-R1-Distill-1.5B) and Qwen3-4B-Instruct-2507 (Yang et al., 2025) (abbreviated as Qwen3-4B-Instruct). Additional results on prompt-selection baselines, code and logic tasks, larger models, other datasets, and policy optimization algorithms are reported in Appendix E.4.

In each iteration, we use a mini-batch of $M = 256$ prompts and a default rollout-count budget of $N_{\text{total}} = M \times 8$. Homogeneous GRPO assigns $N = 8$ rollouts to every prompt, while Knapsack-GRPO keeps the same total budget and varies $N_i$ across prompts. Our models are trained for 1,000 iterations. GRPO and Knapsack-GRPO require comparable total training time: approximately 1,400 GPU hours on A100s for Qwen2.5-Math-7B, or 1 day and 20 hours of wall-clock time with 32 GPUs. This is because the knapsack optimization adds negligible overhead, typically completing in under 1 second, while the per-iteration time remains dominated by response sampling and model training at approximately 130 seconds. Appendix D reports additional step-time and token-length measurements.

For evaluation, we follow (Luo et al., 2025b) and assess our method on several mathematical reasoning benchmarks: AIME, AMC, MATH, MINERVA, and OLYMPIAD Bench (OLYMPIAD for short). Given AIME's small sample size, we combine its 2024 and 2025 editions into a single dataset, hereafter referred to as AIME. Additionally, we include GPQA (Rein et al., 2023) as an out-of-domain evaluation, which tests scientific reasoning across physics, chemistry, and biology. All reported performance metrics are averaged over 16 generated responses.

We report the evaluation performance in Table 1. Knapsack-GRPO improves the average score for all four model settings while keeping the same rollout-count budget as GRPO. For instance, it improves the average score by 3.8 points for DPSK-R1-Distill-1.5B. The gains are not concentrated in one benchmark: Knapsack-GRPO improves by 6.4 points on AIME for DPSK-R1-Distill-1.5B, 9.1 points on AMC for Qwen3-4B-Base, 5.5 points on GPQA for Qwen3-4B-Instruct, and 6.8 points on AMC for Qwen2.5-Math-7B. This pattern supports the central claim that reallocating rollout jobs can improve the usefulness of the generated training signal without changing the overall rollout-count budget.

### 5.2. Experiments with Different Exploration Budgets

We next vary the total exploration budget to study performance under different compute constraints. The default setting uses $N_{\text{total}} = M \times N = 256 \times 8 = 2048$ rollouts per iteration. We additionally evaluate $N_{\text{total}} = 1024$ and $N_{\text{total}} = 4096$. For vanilla GRPO, these correspond to homogeneous rollouts $N = 4$ and $N = 16$, respectively. Knapsack-GRPO uses the same $N_{\text{total}}$.

*Table 1.* Evaluation performance (`avg@16`) comparison across different models and benchmarks.

| | AIME | AMC | MATH | MINERVA | OLYMPIAD | GPQA | Avg |
|---|---|---|---|---|---|---|---|
| DPSK-R1-Distill-1.5B | 25.3 | 62.1 | 81.4 | 25.8 | 41.7 | 39.1 | 42.9 |
| + GRPO | 27.6 | 71.1 | 84.0 | 27.6 | 46.4 | 36.7 | 45.9 |
| + Knapsack-GRPO | **34.0** | **75.1** | **86.7** | **28.5** | **49.7** | **40.3** | **49.7** |
| Qwen3-4B-Base | 6.6 | 29.9 | 48.0 | 19.4 | 23.1 | 26.4 | 22.9 |
| + GRPO | 20.7 | 56.9 | 80.6 | 31.9 | 44.9 | **46.6** | 43.2 |
| + Knapsack-GRPO | **20.8** | **66.0** | **81.0** | **35.7** | **46.2** | 45.5 | **45.1** |
| Qwen3-4B-Instruct | 47.7 | 82.5 | 92.4 | 35.4 | 61.6 | 43.0 | 58.6 |
| + GRPO | 47.0 | **84.9** | 92.5 | **41.8** | 61.8 | 54.4 | 59.2 |
| + Knapsack-GRPO | **48.2** | 83.1 | 92.5 | 38.2 | **63.5** | **59.9** | **61.9** |
| Qwen2.5-Math-7B | 12.3 | 41.0 | 61.2 | 11.8 | 26.1 | 22.0 | 26.7 |
| + GRPO | 23.9 | 70.6 | 81.7 | 33.6 | 41.9 | 40.8 | 45.2 |
| + Knapsack-GRPO | **24.3** | **77.4** | **83.9** | **34.5** | **44.1** | **43.8** | **47.5** |

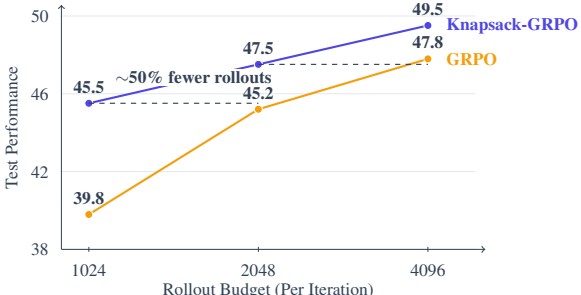

*Figure 4.* Performance comparison under exploration budgets. Knapsack-GRPO with 1024 and 2048 rollouts per iteration achieves performance comparable to GRPO with 2048 and 4096 rollouts, respectively.

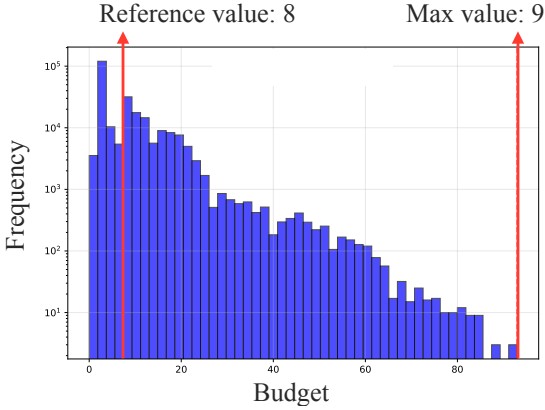

*Figure 5.* Distribution of exploration budgets allocated by Knapsack-GRPO.

Figure 4 shows the Qwen2.5-Math-7B results. Knapsack-GRPO improves the low-budget score from 39.8 to 45.5 and remains stronger at larger budgets. Unlike post-hoc dynamic sampling, which discards uninformative groups and resamples replacements, Knapsack-GRPO proactively assigns heterogeneous rollout counts before generation, moving the same budget away from low-value prompt-budget pairs. Thus, the gain does not rely on significantly more rollout computation: standard GRPO needs about $2\times$ more rollouts for comparable performance.

### 5.3. Understanding Knapsack-based Exploration

We now analyze why knapsack-based allocation improves training, focusing on Qwen2.5-Math-7B. We examine (i) how rollouts are distributed across prompts, (ii) the resulting gradient efficiency, and (iii) changes in prompt difficulty over training.

**Allocated rollout budgets.** We first visualize the rollout counts assigned by Knapsack-GRPO. Figure 5 shows the distribution of allocated budgets at a specific iteration. Even under the same total rollout-count budget, Knapsack-GRPO can assign as many as 93 rollouts to prompts with high marginal learning value, while assigning fewer rollouts (i.e., 2 rollouts) to prompts that are already saturated. In contrast, homogeneous allocation assigns exactly 8 rollouts to every prompt. High-budget prompts can produce longer responses, but Appendix D shows that the measured token-level and runtime overhead remains modest.

**Effective Gradient Ratio.** Figure 6 reports the effective-gradient ratio (EGR; Equation (3)) during training. By reallocating rollouts toward high-marginal-value prompt-budget pairs likely to yield mixed outcomes, Knapsack-GRPO increases EGR by roughly 40% and avoids the monotonic decline under homogeneous allocation. More rollout compute becomes non-zero learning signal, consistent with Section 3. Prompt filtering can further raise EGR; see Appendix E.5.

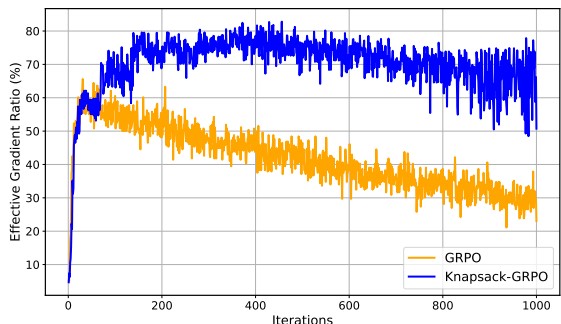

*Figure 6.* Effective gradient ratio during training.

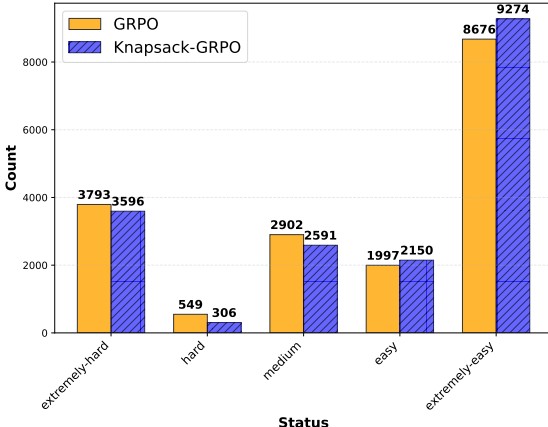

*Figure 7.* Distribution of sample statuses after training.

**Prompt status after training.** Finally, we examine how prompt difficulty evolves. We bucket prompts by success rate $p_i$ into `extremely-hard` ($p_i = 0$), `hard` ($0 < p_i \leq 0.2$), `medium` ($0.2 < p_i < 0.8$), `easy` ($0.8 \leq p_i < 1$), and `extremely-easy` ($p_i = 1$). Figure 7 shows the final distributions. Knapsack-GRPO reduces `extremely-hard` prompts from 3,793 to 3,596 and increases `extremely-easy` prompts from 8,676 to 9,274, suggesting that targeted exploration makes some previously unsolved prompts tractable.

Still, around 20% of prompts remain `extremely-hard`. Training logs show that 577 of them achieve at least one successful trajectory under Knapsack-GRPO, indicating that they are not fundamentally impossible but may require better reuse of rare successes. Incorporating replay or trajectory reuse is a promising direction for future work.

## 6. Related Work

**Resource allocation and knapsack optimization.** Resource allocation is a classical topic in operations research and systems engineering (Hurwicz, 1973; Katoh et al., 2013; Hussain et al., 2013; Maritan & Lee, 2017). Knapsack-style optimization selects high-value decisions under a budget constraint (Pisinger & Toth, 1998). Our work brings this perspective to LLM RL fine-tuning, where rollout generation is the constrained resource. Related theory studies online learning and exploration under knapsack constraints (Badanidiyuru et al., 2018; Chen et al., 2020; Brantley et al., 2020; Li et al., 2021; Liu et al., 2022); our setting focuses on cross-prompt rollout allocation in large-scale RL training.

**Prompt selection and curriculum learning.** A growing line of work improves RL data efficiency through prompt selection, curriculum learning, and difficulty-aware filtering (Lin et al., 2024; Zhang et al., 2024; 2025c; Chen et al., 2025b; Sun et al., 2025a). Recent LLM RL methods use dynamic sampling, online difficulty prediction, selective rollout, rollout replay, or Bayesian task selection to favor prompts likely to yield useful gradients (Yu et al., 2025; Bae et al., 2025; Zheng et al., 2025; Qu et al., 2025; Sun et al., 2025b; Gao et al., 2026; Shen et al., 2026; Mao et al., 2026). These methods decide *which prompts* should be trained on; our work studies *how many rollouts* each selected prompt should receive under a fixed rollout-count budget.

**Efficient RL training systems.** Recent LLM RL systems improve throughput by overlapping rollout generation and model updates, reducing pipeline bubbles, and improving GPU utilization (Fu et al., 2025; Lu et al., 2025; Sheng et al., 2025a). These works address systems efficiency; Knapsack RL targets algorithmic learning efficiency by making the same rollout-count budget produce more useful gradients.

## 7. Conclusion

As RL post-training scales to larger models and longer reasoning trajectories, rollout compute must be used more deliberately. Knapsack RL formulates heterogeneous rollout allocation as a knapsack problem, prioritizing high-value prompt-budget pairs and yielding more effective gradients under the same rollout-count budget. The main lesson is that scaling RL should consider not only how much compute is available, but also how rollout jobs are allocated before generation. This resource-aware view can help make rollout-count budgets more useful for larger models, longer trajectories, and increasingly expensive RL fine-tuning runs.

## Acknowledgements

Tian Ding is supported in part by NSFC (No. 12401409) and Hetao Shenzhen-Hong Kong Science and Technology Innovation Cooperation Zone Project (No. HZQSWS-KCCYB-2024016). Ruoyu Sun is supported by NSFC (No. 12326608), Hetao Shenzhen-Hong Kong Science and Technology Innovation Cooperation Zone Project (No. HZQSWS-KCCYB-2024016), and Guangdong Provincial Key Laboratory of Mathematical Foundations for Artificial Intelligence (No. 2023B1212010001). The work of Z.-Q. Luo was supported by the Guangdong Major Project of

Basic and Applied Basic Research (No. 2023B0303000001), the Guangdong Provincial Key Laboratory of Big Data Computing, and the National Key Research and Development Project under grant 2022YFA1003900.

## Impact Statement

This work focuses on algorithmic design for allocating exploration budgets in RL training for language models. It is purely computational and does not involve human subjects, sensitive data, or ethically contentious datasets. By improving training efficacy, our method may reduce computational cost and the associated carbon footprint of large-scale model development.

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

# A. Additional Related Work

**Data heterogeneity.** A central challenge in RL for LLMs arises from the heterogeneity of training data. Prompts vary substantially in difficulty, leading to diverse reward distributions, and these distributions evolve throughout training, further complicating learning. Prior work has recognized this issue: for example, Li et al. (2024) observed substantial variations in reward distributions across prompts, which complicated stable gradient estimation. Their solution introduced refined baselines to reduce variance, thereby improving the *exploitation* stage of RL. Following this, many advanced policy optimization methods have been proposed (e.g., (Shao et al., 2024; Ahmadian et al., 2024; Yu et al., 2025)). We refer readers to surveys (Zhang et al., 2025b; Wang et al., 2025a) for a broader overview. By contrast, our work directly tackles the *exploration* challenge posed by heterogeneous data, focusing on how to allocate rollout compute across prompts to capture informative trajectories in the first place.

**Scaling RL.** Our method resonates with the principle of test-time scaling (Snell et al., 2024; Brown et al., 2024), which allocates additional computational resources (e.g., best-of-$N$ sampling, majority voting) to improve response quality. In training, Knapsack RL instead asks how to allocate a fixed rollout-count budget so that collected trajectories provide more useful learning signal. More broadly, our work aligns with recent efforts that scale computational budgets in post-training to unlock stronger downstream performance (Jaech et al., 2024; Liu et al., 2025b), as also reflected in recent agentic model releases that expose latency- and cost-aware inference modes (ByteDance Seed, 2026).

**Exploration Strategy.** Our study focuses on simple on-policy exploration with independently sampled rollouts, primarily for its tractability and suitability for global rollout-budget reallocation. However, some difficult tasks may remain unsolved under independent sampling, motivating techniques that improve *how to explore within a fixed prompt*. Tree-based methods, inspired by Monte Carlo Tree Search (e.g., Silver et al., 2016), offer one such direction. Within LLMs, strategies such as state rollbacks and tree-structured reasoning have shown promise (Hou et al., 2025; Li et al., 2025a). We view integrating such within-prompt exploration enhancements with our cross-prompt rollout allocation as a fruitful direction.

Additionally, exploration quality is tightly coupled to output diversity and policy entropy. In supervised fine-tuning, Li et al. (2025b) show that preserving output diversity can improve downstream sampling and test-time compute scaling. In RL, Gao et al. (2025) introduce intrinsic-motivation-guided exploration signals that encourage diverse reasoning trajectories, while Cui et al. (2025b) analyze entropy collapse in reasoning RL and propose regularization techniques to stabilize entropy. These developments are orthogonal to our contribution: such methods improve within-prompt exploration, whereas our work determines how to distribute rollout compute across prompts. Both dimensions can be jointly leveraged.

Finally, we notice that Yao et al. (2025) investigate resource allocation in the context of rejection sampling and RAFT (Dong et al., 2023), focusing on variance reduction. Their work differs from ours in two key ways: (i) it operates outside online RL, and (ii) it does not formulate rollout-budget allocation as a multi-task knapsack problem balancing cost and learning value. Other studies such as Zhang et al. (2025a); Wang et al. (2025b) examine compute allocation during inference, whereas our focus is specifically on optimizing rollout compute during *training*, when rollouts are actively generated and exploration quality is paramount.

# B. Proof

### B.1. Proof of Proposition 1

*Proof of Proposition 1.* We prove both parts of the proposition.

**Part 1: High probability bound.** We want to find a sufficient condition on $N_i$ such that $\mathbb{P}(g_i \neq 0) \geq \alpha$ for a given $\alpha \in (0, 1)$. From the problem setup, we have:
$$\mathbb{P}(g_i \neq 0) = 1 - p^{N_i} - (1-p)^{N_i}.$$
For the condition $\mathbb{P}(g_i \neq 0) \geq \alpha$ to hold, we require:
$$1 - p^{N_i} - (1-p)^{N_i} \geq \alpha$$
$$p^{N_i} + (1-p)^{N_i} \leq 1 - \alpha.$$
Let $q = \max\{p, 1-p\}$. Since $p \in (0, 1)$, we have $q \in [1/2, 1)$, and both $p \leq q$ and $1 - p \leq q$. Hence
$$p^{N_i} + (1-p)^{N_i} \leq 2q^{N_i}.$$

Therefore, it is sufficient to require $2q^{N_i} \leq 1 - \alpha$. Since $\ln q < 0$, this condition is equivalent to

$$\boxed{N_i \geq \frac{\ln\big((1-\alpha)/2\big)}{\ln q} = \frac{\ln\big((1-\alpha)/2\big)}{\ln(\max\{p, 1-p\})}.}$$

This matches the high-probability sufficient condition in Proposition 1.

**Part 2: Expected number of rollouts.** Let $X_1, X_2, \ldots$ be i.i.d. Bernoulli random variables with $\Pr(X_i = 1) = p \in (0, 1)$, where 1 denotes "success" and 0 denotes "failure". Define

$$N^{\text{first}} \equiv N = \min\{n \geq 1 : \text{both } 0 \text{ and } 1 \text{ have appeared among } X_1, \ldots, X_n\}.$$

We compute $\mathbb{E}[N]$ by conditioning on the first trial $X_1$.

**Case 1:** $X_1 = 1$ **(probability $p$).** After the first success, we still need to wait until the first failure occurs. The waiting time for the first failure follows a geometric distribution with success probability $1 - p$, whose expectation is $1/(1 - p)$. Thus

$$\mathbb{E}[N \mid X_1 = 1] = 1 + \frac{1}{1 - p}.$$

**Case 2:** $X_1 = 0$ **(probability $1 - p$).** By symmetry, we wait for the first success; its waiting time has expectation $1/p$, so

$$\mathbb{E}[N \mid X_1 = 0] = 1 + \frac{1}{p}.$$

Applying the law of total expectation:

$$\mathbb{E}[N] = p\Big(1 + \frac{1}{1 - p}\Big) + (1 - p)\Big(1 + \frac{1}{p}\Big)$$

$$= \frac{1}{p} + \frac{1}{1 - p} - 1.$$

Hence, the expected number of rollouts until we first observe both a success and a failure is

$$\boxed{\mathbb{E}[N^{\text{first}}] = \frac{1}{p} + \frac{1}{1 - p} - 1.}$$

This completes the proof of the second part of Proposition 1. $\square$

### B.2. Proof of Proposition 2

*Proof of Proposition 2.* We use a simplified single-step policy-gradient model to approximate the change in the prompt success rate. The notation $p_y$ below denotes the probability of the successful response mode $y$; in the main text this quantity is identified with the prompt success rate $p_i$. We make the following assumptions:

- The policy follows a softmax distribution: $p_k = \frac{\exp(z_k)}{\sum_{j=1}^{K} \exp(z_j)}$ for action $k$.

- The gradient update follows the policy gradient rule with advantage $A$:

$$z_k \leftarrow z_k + \eta A \nabla_{z_k} \log p_y$$

  where $\eta$ is the learning rate and $y$ is the chosen action.

- We assume unit learning rate ($\eta = 1$) and unit advantage ($A = 1$) for simplicity.

**Step 1: Taylor expansion.** For small parameter changes, the change in success probability can be approximated by:

$$\Delta p_y \approx \sum_{k=1}^{K} \frac{\partial p_y}{\partial z_k} \times \Delta z_k$$

**Step 2: Computing partial derivatives.** For the softmax probability $p_y = \frac{\exp(z_y)}{\sum_{j=1}^{K} \exp(z_j)}$, we have:

$$\frac{\partial p_y}{\partial z_y} = p_y(1 - p_y), \quad \text{and} \quad \frac{\partial p_y}{\partial z_k} = -p_y p_k, \quad \text{for } k \neq y.$$

**Step 3: Determining parameter updates.** Under the policy gradient update rule, we have:

$$\nabla_{z_k} \log p_y = \mathbb{I}[k = y] - p_k.$$

Therefore, the parameter updates are:

$$\Delta z_y = \mathbb{I}[y = y] - p_y = 1 - p_y,$$
$$\Delta z_k = \mathbb{I}[k = y] - p_k = 0 - p_k = -p_k, \quad \text{for } k \neq y$$

**Step 4: Computing InfoGain.** Substituting the partial derivatives and parameter updates:

$$\Delta p_y = \frac{\partial p_y}{\partial z_y} \Delta z_y + \sum_{k \neq y} \frac{\partial p_y}{\partial z_k} \Delta z_k$$

$$= p_y(1 - p_y) \cdot (1 - p_y) + \sum_{k \neq y} (-p_y p_k) \cdot (-p_k)$$

$$= p_y(1 - p_y)^2 + p_y \sum_{k \neq y} p_k^2$$

**Step 5: Simplification under a sparse-success approximation.** In LLM reasoning tasks, the successful response region is usually a small part of a very large response space. Under this abstraction, the cross-logit term $p_y \sum_{k \neq y} p_k^2$ is treated as a higher-order correction relative to the direct effect on the successful response mode. This yields the approximation used in the main text:

$$\text{InfoGain} \approx \boxed{p_y(1 - p_y)^2}.$$

Identifying $p_y$ with $p_i$ gives $\text{InfoGain}(p_i) \approx p_i(1 - p_i)^2$. $\qquad\square$

To validate this approximation, we conduct an empirical study with 100 actions, comparing the `InfoGain` computed through exact gradient updates against our theoretical approximation from Proposition 2. As shown in Figure 8, the two curves align closely across different success rates, demonstrating that our formula $p(1 - p)^2$ provides a reliable approximation for practical use.

We also evaluate the qualitative prediction in a more LLM-like setting. Specifically, we fine-tune Qwen2.5-Math-7B on single prompts using positive examples and then measure the resulting improvement on the same prompt. We group prompts by their initial success-rate bins and compare the empirical improvement with the theoretical score $p(1 - p)^2$. As shown in Table 2, the empirical improvements concentrate in moderately difficult bins and decrease near the extremes, matching the qualitative shape of the approximation even though the assumptions used in the Taylor analysis are simplified.

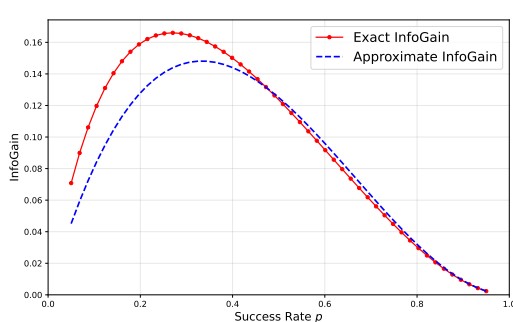

*Figure 8.* Comparison of exact `InfoGain` and approximate formula.

*Table 2.* Empirical validation of the InfoGain approximation in a Qwen2.5-Math-7B single-prompt fine-tuning study.

| Success Rate Bin | 0.1 | 0.2 | 0.3 | 0.4 | 0.5 | 0.6 | 0.7 | 0.8 | 0.9 |
|---|---|---|---|---|---|---|---|---|---|
| Empirical measurement | 0.06 | 0.13 | 0.22 | 0.10 | 0.14 | 0.06 | 0.05 | 0.07 | 0.03 |
| Theoretical prediction | 0.08 | 0.13 | 0.15 | 0.14 | 0.13 | 0.10 | 0.06 | 0.03 | 0.01 |

**B.3. Proof of Proposition 3**

*Proof of Proposition 3.* To simplify notations, we write

$$\text{Value}(N_i, p_i) \equiv V(N_i; p_i) := \left(1 - p_i^{N_i} - (1 - p_i)^{N_i}\right) p_i(1 - p_i)^2$$

and

$$c(p) := p(1-p)^2 > 0, \qquad f(N;p) := 1 - p^N - (1-p)^N,$$

so that $V(N;p) = f(N;p)\,c(p)$.

**(1) Monotonicity.** We have

$$f(N+1;p) - f(N;p) = \left(1 - p^{N+1} - (1-p)^{N+1}\right) - \left(1 - p^N - (1-p)^N\right)$$
$$= p^N(1-p) + (1-p)^N p > 0,$$

since $p \in (0,1)$. Multiplying by $c(p) > 0$ shows

$$\Delta V(N;p) = V(N+1;p) - V(N;p) > 0,$$

so $V(N;p)$ is strictly increasing (hence non-decreasing) in $N$.

**(2) Diminishing returns.** Using the above expression,

$$\Delta V(N;p) = \left[p^N(1-p) + (1-p)^N p\right] c(p).$$

The second discrete difference is

$$\Delta^2 V(N;p) := \Delta V(N+1;p) - \Delta V(N;p)$$
$$= \left[p^{N+1}(1-p) + (1-p)^{N+1}p - p^N(1-p) - (1-p)^N p\right] c(p).$$

Factor the bracketed term:

$$p^{N+1}(1-p) - p^N(1-p) = p^N(1-p)(p-1) = -p^N(1-p)^2,$$
$$(1-p)^{N+1}p - (1-p)^N p = (1-p)^N p\big((1-p) - 1\big) = -(1-p)^N p^2.$$

Therefore,

$$\Delta^2 V(N;p) = \left[-p^N(1-p)^2 - (1-p)^N p^2\right] c(p) < 0,$$

since each factor is positive and the sum inside the brackets is negative. Hence $\Delta V(N;p)$ is strictly decreasing in $N$, establishing diminishing returns.

**(3) Hard-task bias.** Observe that $f(N;p)$ is symmetric in $p$ and $1-p$,

$$f(N;p) = 1 - p^N - (1-p)^N = f(N; 1-p),$$

while the multiplicative factor $c(p)$ is not:

$$c(p) = p(1-p)^2, \qquad c(1-p) = (1-p)p^2.$$

Thus

$$\frac{V(N;p)}{V(N;1-p)} = \frac{f(N;p)c(p)}{f(N;1-p)c(1-p)} = \frac{c(p)}{c(1-p)} = \frac{p(1-p)^2}{(1-p)p^2} = \frac{1-p}{p}.$$

For $p \in (0, 1/2)$, we have $(1-p)/p > 1$, so $V(N;p) > V(N;1-p)$.

An analogous argument applies to the marginal values. From the expression above,

$$\Delta V(N;p) = \left[p^N(1-p) + (1-p)^N p\right] c(p),$$

and the bracketed term is again symmetric in $p$ and $1-p$. Therefore

$$\frac{\Delta V(N;p)}{\Delta V(N;1-p)} = \frac{c(p)}{c(1-p)} = \frac{1-p}{p} > 1 \quad \text{for } p \in (0, 1/2),$$

which implies $\Delta V(N;p) > \Delta V(N;1-p)$ for all $N \geq 1$. This completes the proof. $\qquad\square$

### B.4. Proof of Proposition 4

We introduce the formal version of Proposition 4 below.

**Theorem 1** (Greedy optimal allocation). *Assume* $\mathrm{Value}(N;p) \equiv V(N;p)$ *is given as in Equation* (9). *Furthermore, let* $\Delta V_i(N_i)$ *be the marginal value of $i$-th task. An optimal allocation $\{N_i^\star\}$ for Equation* (5) *can be obtained by the following*

*greedy / water-filling procedure:*

- *Initialize $N_i \leftarrow N_{\text{low}}$ for all $i$.*

- *While $\sum_{i=1}^{M} N_i < N_{\text{total}}$:*
  - *Select an index*
  
  $$j \in \arg \max_{i:N_i < N_{\text{up}}} \Delta V_i(N_i),$$
  
  *i.e., the task with the largest current marginal gain that has not yet hit its upper bound.*
  - *Update $N_j \leftarrow N_j + 1$.*

*The resulting allocation $\{N_i\}$ is an optimal solution of Equation (5).*

*Proof of Theorem 1.* By Proposition 3(2), for each task $i$ the sequence $N \mapsto V(N; p_i)$ is discrete concave, so its marginal gains $\Delta V_i(N)$ are strictly decreasing in $N$. Consider an arbitrary feasible allocation $\{N_i\}$, and imagine constructing it by "adding" one trajectory at a time starting from the lower bounds.

Each trajectory addition to task $i$ corresponds to selecting one element from the decreasing sequence

$$\Delta V_i(N_{\text{low}}), \ \Delta V_i(N_{\text{low}} + 1), \ \ldots, \ \Delta V_i(N_{\text{up}} - 1).$$

The total objective value can therefore be written as the sum of exactly $N_{\text{total}} - MN_{\text{low}}$ chosen marginal gains across all tasks.

The greedy algorithm selects these marginal gains in non-increasing order, subject to the per-task capacity constraints (at most $N_{\text{up}} - N_{\text{low}}$ gains can be taken from each sequence). This is optimal because of a standard exchange argument: if some feasible allocation uses a marginal gain $\delta_a$ that is strictly smaller than another available marginal gain $\delta_b$ (which respects the same capacity constraints) that it did not use, then swapping $\delta_a$ for $\delta_b$ strictly increases the objective while preserving feasibility. By repeatedly applying such exchanges, any non-greedy allocation can be transformed into the greedy one without decreasing the objective, so the greedy allocation is optimal.

Formally, one can view the multiset of all candidate marginal gains

$$\left\{ \Delta V_i(N_{\text{low}}), \ldots, \Delta V_i(N_{\text{up}} - 1) \right\}_{i=1}^{M}$$

as a collection of sorted sequences; the greedy algorithm chooses the globally largest feasible entries, and any deviation from this choice admits an improving swap. This establishes the optimality of the greedy procedure. $\square$

## C. Extensions

In this work, we mainly focus on the widely used GRPO (Shao et al., 2024) algorithm to design the optimal allocation strategy. Here we discuss possible extensions for other RL algorithms by adapting the core framework while maintaining the same task value function structure:

$$\text{Value}(N_i, p_i) = \text{ProbNonZeroGradient}(N_i, p_i) \times \text{InfoGain}(p_i).$$

The key difference lies in how we compute $\text{ProbNonZeroGradient}(N_i, p_i)$ for different algorithms:

- **RLOO** (Ahmadian et al., 2024). RLOO's policy gradient estimator is equivalent to GRPO up to constants, thus we may not need fundamental changes. The probability of obtaining a non-zero gradient remains:

  $$\text{ProbNonZeroGradient}(N_i, p_i) = 1 - p_i^{N_i} - (1 - p_i)^{N_i}.$$

- **ReMax** (Li et al., 2024). ReMax leverages the reward of greedy response as baseline, rather than the averaged reward used in GRPO. In this setting, a gradient update occurs only when the sampled trajectory differs from the greedy response. If we denote the probability of the greedy response as $\alpha$, then the probability of sampling a trajectory different from the greedy response is $1 - \alpha$. The probability of obtaining a non-zero gradient with $N_i$ samples becomes:

  $$\text{ProbNonZeroGradient}(N_i, \alpha) = 1 - \alpha^{N_i}.$$

  This represents the probability that at least one of the $N_i$ sampled trajectories differs from the greedy response, thereby producing a gradient signal.

- **REINFORCE** (Williams, 1992). There is no baseline design in vanilla REINFORCE. We can directly calculate the ProbNonZeroGradient to account for the case where at least one trajectory receives a positive reward:

$$\text{ProbNonZeroGradient}(N_i, p_i) = 1 - (1 - p_i)^{N_i}.$$

This formulation is simpler than GRPO since we only need to ensure at least one successful trajectory occurs, rather than balancing positive and negative samples.

The proposed framework's modularity allows for straightforward adaptation to other RL algorithms by: (1) identifying the algorithm's gradient computation mechanism, (2) determining conditions for non-zero gradients, (3) calculating the corresponding $\text{ProbNonZeroGradient}$ function, and (4) maintaining the same $\text{InfoGain}(p_i) = p_i(1 - p_i)^2$ formulation across algorithms. This demonstrates the general applicability of our value-based budget allocation approach beyond the specific GRPO implementation.

**Extension to Continuous-Reward RLHF.** The binary-reward formulation used in RLVR gives a clean notion of zero-gradient groups: all-positive and all-negative groups both produce zero relative advantages under GRPO. Standard RLHF with reward-model scores is different because rewards are continuous and the exact event of all rollouts receiving identical rewards is unlikely. In this setting, the knapsack abstraction can still be used, but the task value should be redefined around gradient signal quality rather than binary success/failure.

One natural replacement is an advantage-variance or gradient signal-to-noise objective. For a prompt $x_i$, let $A_{i,j}$ denote the normalized advantage of rollout $j$. Instead of $P_{\text{nz}}(N_i, p_i)$, one can use a term that measures whether the rollout group is expected to produce meaningful advantage variation, such as $\text{Var}(A_i)$ or the expected squared norm of the prompt-level policy-gradient estimate. The value of additional rollouts can then be modeled as improving the signal-to-noise ratio of

$$\widehat{g}_i = \frac{1}{N_i} \sum_{j=1}^{N_i} A_{i,j} \nabla_\theta \log \pi_\theta(y_{i,j} \mid x_i),$$

where the estimation noise decreases with $N_i$. Prompts with nearly constant reward-model scores have low advantage variance and receive little additional value, while prompts with high reward uncertainty may benefit from more samples. This continuous-reward variant is outside the empirical scope of this paper, but it illustrates that the resource-allocation framework is not tied to binary rewards; what changes is the value model used by the knapsack solver.

## D. Experiment Details

Our experiments utilized the large-scale RL training framework `Verl`, specifically version 0.5.0. No modifications were made to the core training and inference code, with the exception of the advantage calculation, where values were clipped between -5 and 5. This was implemented because, as rollout responses were scaled, we observed their values could become significantly large in extreme cases, thus requiring this additional clipping for numerical stability. Additional implementation details on handling extreme cases and ensuring rollout efficiency are provided in this appendix.

Following recommendations from (Yu et al., 2025), the learning rate was set to $10^{-6}$, with importance sampling clipping ratios (high/low) of 0.28 and 0.2, respectively. Neither KL nor entropy regularization was employed. Models were trained with a maximum sequence length of 4K tokens, with the exception of DPSK-R1-Distill-1.5B, which utilized 8K tokens to accommodate its typically longer chain-of-thought behaviors requiring more context.

For knapsack-based exploration, we use the success rate from rollouts in the last epoch, calculated empirically. Since our framework may assign a small budget (e.g., 2) to certain prompts, this can make the success rate estimation sensitive. To address this, we incorporate rollouts from earlier epochs, managing a buffer of the last 16 samples to smooth the success rate estimation. Note that our value function defined in Section 4.3 assigns a zero value to prompts with empirical success rates of 0 or 1, which would otherwise lead to zero budget allocation for these prompts. To prevent their complete exclusion and maintain coverage:

- For $\widehat{p}_i = 1.0$ (prompts always solved correctly), the estimate may be not accurate from history samples, so we allocate a small minimum budget (e.g., 2) to ensure they are still considered. This can be achieved by set $N_{\text{low}}$ in Equation (5).

- For $\widehat{p}_i = 0.0$ (prompts never solved correctly), we employ a fallback allocation strategy. We first estimate the total budget required for prompts with $p_i \in (0, 1]$ according to Proposition 1 and the above rule. Any remaining budget is subsequently distributed among extremely hard tasks. This strategy is particularly beneficial in later training stages where many prompts

become easy, thus freeing up capacity to focus on hard tasks.

For evaluation results reported during training, models were assessed every 10 training iterations using 16 generated responses. To manage evaluation time, 100 evaluation samples were randomly selected from benchmarks when the total number of samples exceeded this number.

For the final evaluation performance presented in Table 1, different maximum sequence lengths were used to prevent response truncation: 4K tokens for Qwen2.5-Math-7B, 8K tokens for Qwen3-4B and Qwen3-4B-Instruct, and 16K tokens for DPSK-R1-Distill-1.5B. Consequently, these results may not perfectly align with those reported in the training curves.

We conducted experiments using mixed computational resources: Qwen2.5-Math-7B was trained on $32\times$A100-80GB GPUs for 1 day and 20 hours; DPSK-R1-Distill-1.5B on $32\times$H20-96GB GPUs for 1 day and 10 hours; Qwen3-4B on $32\times$H20-96GB GPUs for 1 day and 5 hours; and Qwen3-4B-Instruct on the same hardware for 2 days and 16 hours. We note that the knapsack optimization itself is computationally negligible, typically taking less than 1 second.

**Runtime and Token-Level Cost.** Our main formulation treats each rollout as one unit of cost. This is a rollout-count approximation: in practice, prompts receiving larger budgets can also induce longer generations. For example, the average generated length of high-budget prompts is higher than that of low-budget prompts (1,135 vs. 953 tokens for Qwen2.5-Math-7B, and 6,614 vs. 4,556 tokens for DPSK-R1-Distill-1.5B). We therefore keep the efficiency claim at the rollout-count level and report wall-clock measurements in Table 3. For Qwen2.5-Math-7B, Knapsack-GRPO has similar rollout time and only a small increase in average step time. For the longer-CoT DPSK-R1-Distill-1.5B model, the average step-time overhead is below 4.2%. These results suggest that heterogeneous allocation improves the use of rollout-count budget, while its actual token-level cost is slightly higher when high-budget prompts also produce longer responses.

*Table 3.* Wall-clock comparison between GRPO and Knapsack-GRPO. Step time includes rollout generation, reward computation, and model update.

| Method | Avg Step Time (s) | Rollout Time (s) |
|---|---|---|
| Qwen2.5-Math-7B + GRPO | 110 | 43 |
| Qwen2.5-Math-7B + Knapsack-GRPO | 115 | 40 |
| DPSK-R1-Distill-1.5B + GRPO | 239 | 112 |
| DPSK-R1-Distill-1.5B + Knapsack-GRPO | 249 | 120 |

# E. Additional Results

## E.1. Visualization of Exploration Process

**Evolution of Prompts.** To illustrate the impact of exploration budgets on individual prompt learning dynamics, we track and visualize the learning trajectories of several randomly selected prompts from the training data in Figure 9. Each subplot corresponds to a unique prompt, identified by its index in the title. We observe that for several examples, our framework effectively allocates more exploration budget, leading to complete learning of the prompt (e.g., prompts in the first row, first column, and second row, first column). Conversely, some tasks remain highly challenging, where neither Knapsack-GRPO nor GRPO achieves satisfactory performance (e.g., the prompt in the third row, second column).

## E.2. Training Curves.

As references, the training curves for all models are displayed in Figures 10, 11, 12, and 13. Compared with the final results in Table 1, these plots further show that Knapsack-GRPO delivers a rapid performance improvement early in the training process. We also observe a few cases of performance degeneration, which points to the need for exploring more stable policy optimization techniques in future research.

## E.3. Success Rate Estimation

We demonstrate the effectiveness of the success rate estimation strategy used in our work. Instead of relying on the success rate estimated from previous rollouts, we generate fresh rollouts (8 rollouts per prompt) at each step and estimate the success

---

**Algorithm 1** Knapsack GRPO

---

**Input:** Model $\pi_\theta$, reward $r(x, y)$, total budget $N_{\text{total}} = N \times M$, bounds $N_{\text{low}}, N_{\text{up}}$, initial success-rate map $\widehat{p}_0(\cdot)$ (optional)

1: **Initialize success rates:**
2: **if** user provides $\widehat{p}_0(\cdot)$ **then**
3:    $\widehat{p}(\cdot) \leftarrow \widehat{p}_0(\cdot)$
4: **else**
5:    $\widehat{p}(x) \leftarrow$ `empty` for all prompts $x$
6: **end if**
7: **for** iteration $t = 1, 2, \ldots$ **do**
8:    Sample a mini-batch of prompts $\{x_1, \ldots, x_M\}$
9:    **— Budget Allocation —**
10:    **if** there exists $\widehat{p}(x_i)$ is *empty* **then**
11:      Set $N_i \leftarrow N$ for all $i$
12:    **else**
13:      Compute $\{N_1, \ldots, N_M\}$ by solving the knapsack problem Equation (5) using $\widehat{p}(x_i)$
14:    **end if**
15:    **— Execution —**
16:    Generate $N_i$ responses $\{y_{i,1}, \ldots, y_{i,N_i}\} \sim \pi_\theta(\cdot|x_i)$ for each prompt $x_i$
17:    **— Optimization (GRPO) —**
18:    Compute GRPO gradient $g(\theta)$ via Equation (2)
19:    Update model parameters: $\theta \leftarrow \theta + \eta\, g(\theta)$
20:    **— Success-Rate Update —**
21:    **for** $i = 1, \ldots, M$ **do**
22:      Update $\widehat{p}(x_i)$ using the observed rewards $\{r(x_i, y_{i,1}), \ldots, r(x_i, y_{i,N_i})\}$
23:    **end for**
24: **end for**

---

rate before each budget allocation.[1] As shown in Figure 14, we measure the error using the absolute difference between the success rate estimates. Initially, the error is around 0.25, but it decreases to 0.08 by the end of training. This trend is primarily due to the fact that the model parameters change slowly during training, allowing us to leverage past rollouts to more accurately estimate the current success rate. Given the small magnitude of this error, we expect it to have a minimal impact on the budget allocation.

Note that we also evaluate performance using this improved estimation strategy, which resulted in a final performance of 47.7, slightly better than the 47.5 achieved with our original strategy. However, this approach introduces a 2x increase in generation time, leading to slower training.

We also have explored an online logistic regression method to predict success rates. Using a dataset of 10,000 samples, we train a 2-layer MLP on prompt embeddings from the Qwen3-0.6B-Embedding model. This predictor achieves an error of approximately 0.18 on a separate evaluation dataset of 256 samples, demonstrating the promise of this approach. Fully integrating this predictive model into the online training infrastructure remains a direction for future work.

### E.4. Ablation Studies

**Results with Llama Models.** We provide additional results using the Llama-3.1-8B-Instruct model. For this model, we fine-tune with the `MATH` dataset (Hendrycks et al., 2021) instead of `DAPO-MATH-17K`, as the latter proves too challenging for Llama-3.1-8B-Instruct. Additionally, we disable off-policy updates, as we observe they cause training instability for this model over extended training iterations. The results are reported in Table 4. We observe that Knapsack-GRPO achieves the best overall performance, improving the average score from 18.7 (base model) to 24.5, outperforming standard GRPO (22.6) across most benchmarks, with particularly notable gains on AMC (29.1 vs. 23.6) and GPQA (38.5 vs. 34.9).

---

[1]While this approach eliminates the delay issue, it still cannot fully eliminate the error introduced by the finite number of samples.

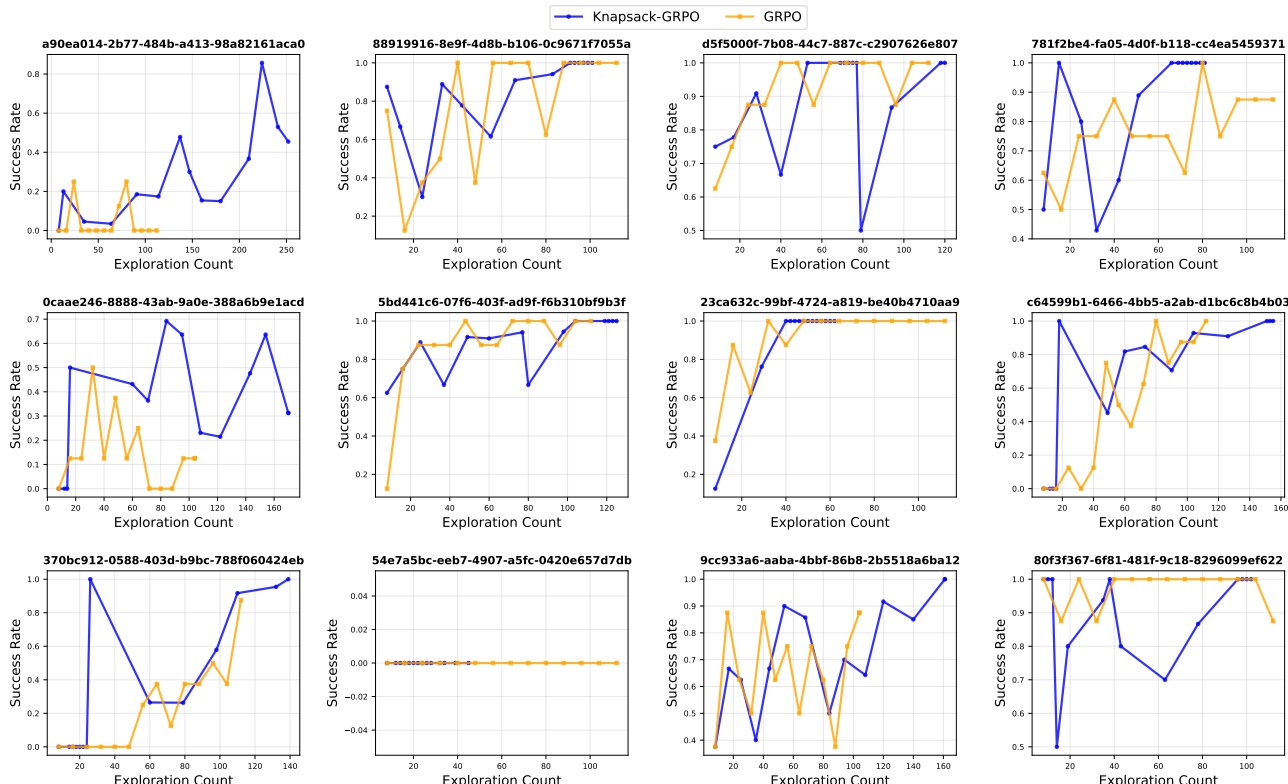

*Figure 9.* Learning dynamics of randomly selected prompts throughout training, comparing GRPO and Knapsack-GRPO. Each subplot shows the success rate evolution for a specific prompt.

*Table 4.* Evaluation performance (`avg@16`) with Llama3.1-8B-Instruct model.

|  | AIME | AMC | MATH | MINERVA | OLYMPIAD | GPQA | Avg |
|---|---|---|---|---|---|---|---|
| Llama3.1-8B-Instruct | 2.1 | 19.3 | 47.6 | 17.3 | 15.2 | 27.2 | 18.7 |
| + GRPO | **4.3** | 23.6 | 51.0 | 21.9 | 18.4 | 34.9 | 22.6 |
| + Knapsack-GRPO | 4.1 | **29.1** | **53.2** | **23.6** | **18.7** | **38.5** | **24.5** |

**Results with Qwen2.5-14B.** To examine whether the method scales beyond 7B models, we additionally train Qwen2.5-14B-Instruct on DAPO-Math-17K. As shown in Table 5, Knapsack-GRPO improves the average score from 45.9 to 47.4 and outperforms GRPO on five of six benchmarks. The effective-gradient ratio also increases from 34.0% to 62.2%, suggesting that heterogeneous rollout allocation remains useful at a larger model scale.

**Results on Code and Logic Tasks.** Beyond mathematical reasoning, we also evaluate whether heterogeneous rollout allocation transfers to other RLVR domains. For code generation, we train Qwen2.5-7B on the code split from PRIME-RL (Cui et al., 2025a). For logical reasoning, we train Qwen2.5-7B on the easy split from SynLogic (Liu et al., 2025a). Results are shown in Table 6. Knapsack-GRPO improves the average score by 1.5 points on code tasks and 4.9 points on logic tasks. On SynLogic, the effective-gradient ratio increases by roughly 50%, consistent with our main diagnosis.

**Results with DeepScaleR Dataset** Beyond the `DAPO-MATH-17K` dataset, we also consider the `DeepScaleR` training dataset (Luo et al., 2025b), which contains 40,315 training queries with verifiable ground truths. We fine-tune the Qwen2.5-Math-7B model on this dataset; results are shown in Table 7. We see that Knapsack-GRPO consistently outperforms standard GRPO, achieving an average score of 45.6 compared with 43.9. The improvements are particularly pronounced on AIME

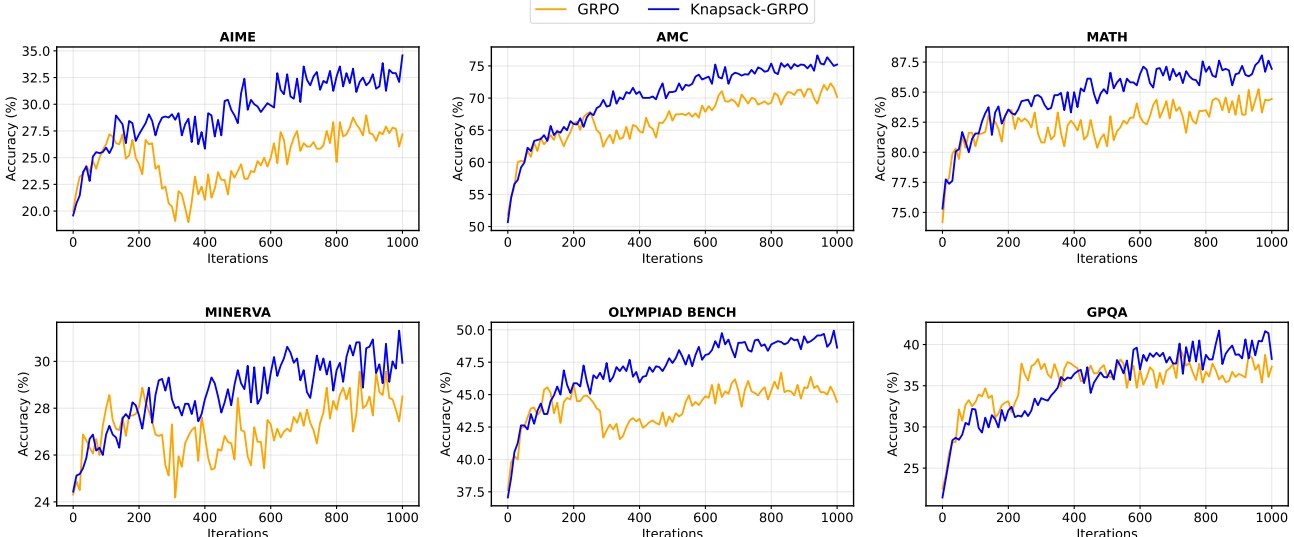

*Figure 10.* Evaluation performance of DPSK-R1-Distill-1.5B across training iterations.

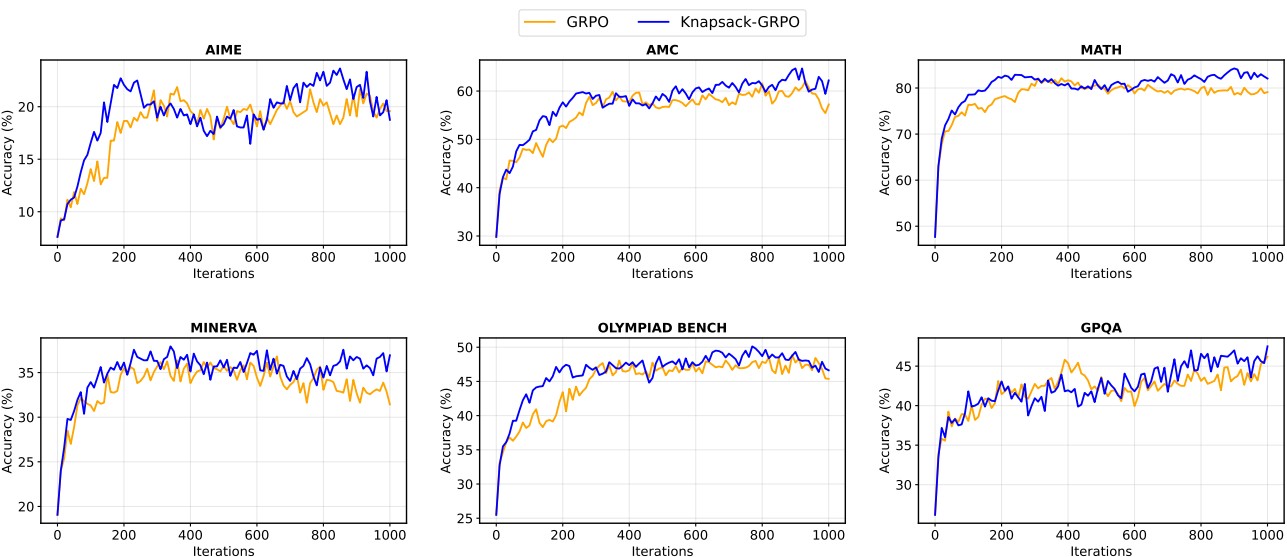

*Figure 11.* Evaluation performance of Qwen3-4B-Base across training iterations.

*Table 5.* Evaluation performance (`avg@16`) with Qwen2.5-14B-Instruct.

| | AIME | AMC | MATH | MINERVA | OLYMPIAD | GPQA | Avg |
|---|---|---|---|---|---|---|---|
| Qwen2.5-14B-Instruct | 12.0 | 50.5 | 75.0 | 32.8 | 41.4 | 40.9 | 42.1 |
| + GRPO | 14.8 | 59.1 | 76.4 | 35.2 | **46.8** | 43.0 | 45.9 |
| + Knapsack-GRPO | **16.4** | **62.0** | **78.8** | **36.4** | 45.9 | **44.8** | **47.4** |

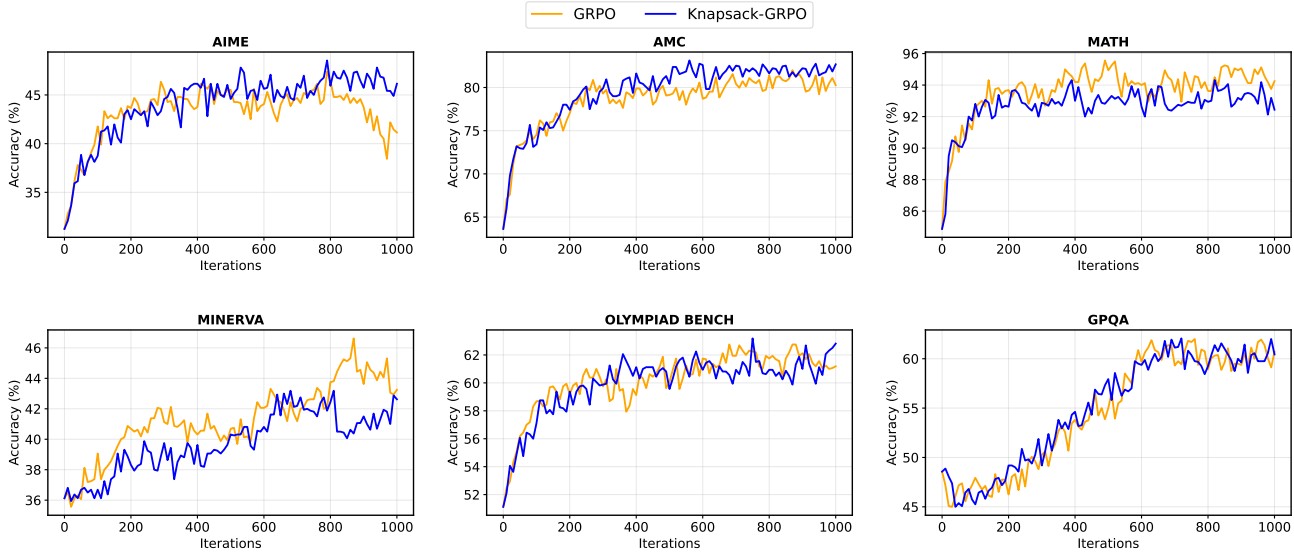

*Figure 12.* Evaluation performance of Qwen3-4B-Instruct across training iterations.

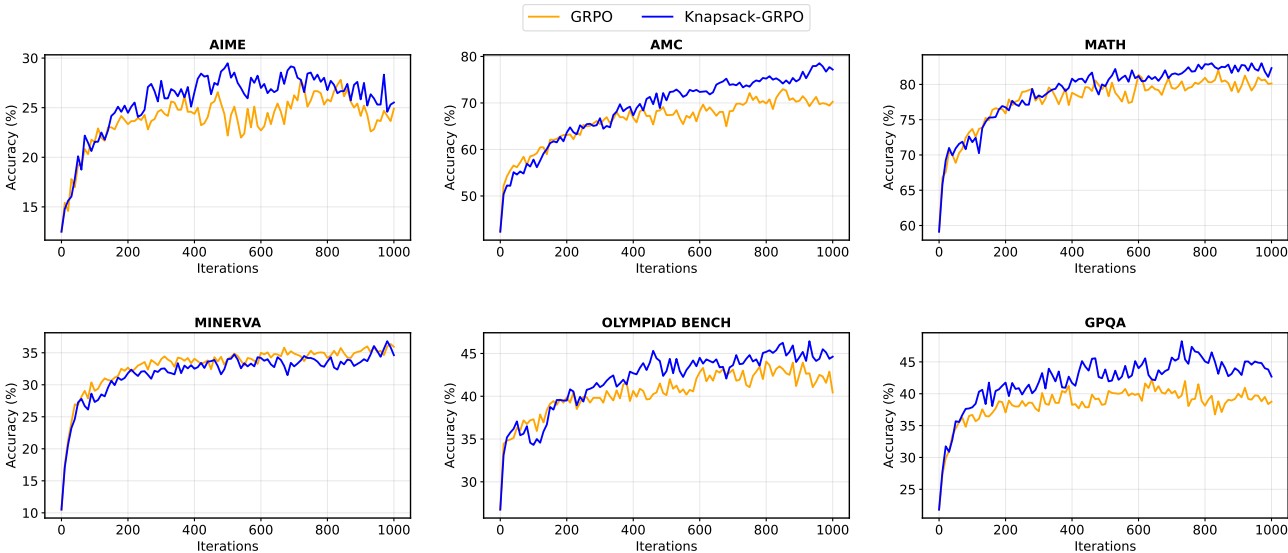

*Figure 13.* Evaluation performance of Qwen2.5-Math-7B across training iterations.

*Table 6.* Generalization to code and logic RLVR tasks with Qwen2.5-7B.

| Code Generation | | | | | |
|---|---|---|---|---|---|
| **Method** | **Taco** | **Codeforces** | **CodeContests** | **APPS** | **LiveCodeBench** | **Avg** |
| GRPO | 18.0 | **21.3** | 31.0 | 32.8 | **22.8** | 24.8 |
| Knapsack-GRPO | **23.0** | 20.3 | **34.0** | **36.0** | 22.3 | **26.3** |

| Logic Tasks | | | | | |
|---|---|---|---|---|---|
| **Method** | **Arc** | **Agicalc** | **Math_path** | **Sudoku** | **Web_of_lies** | **Avg** |
| GRPO | 8.3 | 11.3 | 78.9 | 72.5 | **15.0** | 47.6 |
| Knapsack-GRPO | **22.2** | **15.0** | **86.3** | **85.0** | 12.5 | **52.5** |

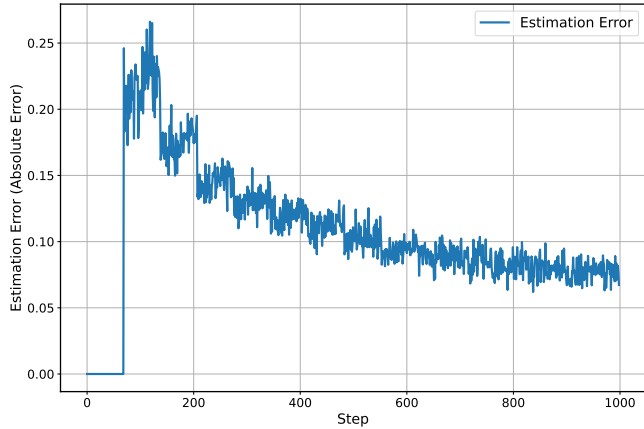

*Figure 14.* Error in success rate estimation over the course of training, measured using the total variation distance.

(28.4 vs. 25.0), MINERVA (37.1 vs. 34.7), OLYMPIAD (44.4 vs. 42.9), and GPQA (34.3 vs. 32.6), demonstrating the effectiveness of our approach across diverse mathematical reasoning benchmarks.

*Table 7.* Evaluation performance (`avg@16`) with `DeepScaleR` dataset.

| | AIME | AMC | MATH | MINERVA | OLYMPIAD | GPQA | Avg |
|---|---|---|---|---|---|---|---|
| Qwen2.5-Math-7B | 12.3 | 41.0 | 61.2 | 11.8 | 26.1 | 22.0 | 26.7 |
| + GRPO | 25.0 | 63.9 | 83.2 | 34.7 | 42.9 | 32.6 | 43.9 |
| + Knapsack-GRPO | **28.4** | **62.7** | **84.1** | **37.1** | **44.4** | **34.3** | **45.6** |

**Results with REINFORCE++.** Beyond GRPO, we also evaluate our exploration–budget allocation strategy with another RL algorithm, REINFORCE++ (Hu et al., 2025). As shown in Table 8, replacing the homogeneous allocation with our knapsack-based scheme consistently improves performance on most tasks (e.g., +6.6 on MINERVA and +4.5 on OLYMPIAD), while AIME remains comparable. Overall, the average score increases from 39.3 to 41.1.

*Table 8.* Evaluation performance (`avg@16`) with `REINFORCE++` training algorithm.

| | AIME | AMC | MATH | MINERVA | OLYMPIAD | GPQA | Avg |
|---|---|---|---|---|---|---|---|
| Qwen2.5-Math-7B | 12.3 | 41.0 | 61.2 | 11.8 | 26.1 | 22.0 | 26.7 |
| + REINFORCE++ | **20.2** | 62.4 | 76.5 | 25.3 | 32.5 | 38.3 | 39.3 |
| + Knapsack-REINFORCE++ | 18.9 | **63.2** | **78.9** | **31.9** | **37.0** | **39.1** | **41.1** |

**Without Fallback Strategy.** In Appendix D, we introduced the *fallback strategy*, which reallocates excess exploration budgets from already-solved prompts to those that remain unsolved. This prevents a common failure mode: difficult prompts may otherwise receive too few resources, while easy prompts are oversampled.

A concrete example is shown in Table 9 with 8 prompts. Without the fallback strategy, the allocation assigns over 50 exploration units to a task with a success rate of 0.9, while the unsolved task (success rate 0.0) receives only 2 units. In contrast, with the fallback strategy, the unsolved task is assigned 29 units—substantially increasing its chance of making progress.

Empirically, this design proves crucial (Figure 15). In our experiments with the Qwen2.5-Math-7B model, removing the fallback strategy led to unstable training, large performance fluctuations on benchmarks such as AMC and OlympiadBench,

*Table 9.* Comparison of budget allocation with and without fallback strategy.

| | With Fallback Strategy | | | Without Fallback Strategy | | |
|---|---|---|---|---|---|---|
| Index | Success Rate | Cost | Assignment | Success Rate | Cost | Assignment |
| 1 | 0.0 | $\infty$ | 29 | 0.0 | $\infty$ | 2 |
| 2 | 0.9 | 22 | 23 | 0.9 | 22 | 50 |
| 3 | 1.0 | 0 | 2 | 1.0 | 0.0 | 2 |
| 4 | 1.0 | 0 | 2 | 1.0 | 0.0 | 2 |
| 5 | 1.0 | 0 | 2 | 1.0 | 0.0 | 2 |
| 6 | 1.0 | 0 | 2 | 1.0 | 0.0 | 2 |
| 7 | 1.0 | 0 | 2 | 1.0 | 0.0 | 2 |
| 8 | 1.0 | 0 | 2 | 1.0 | 0.0 | 2 |

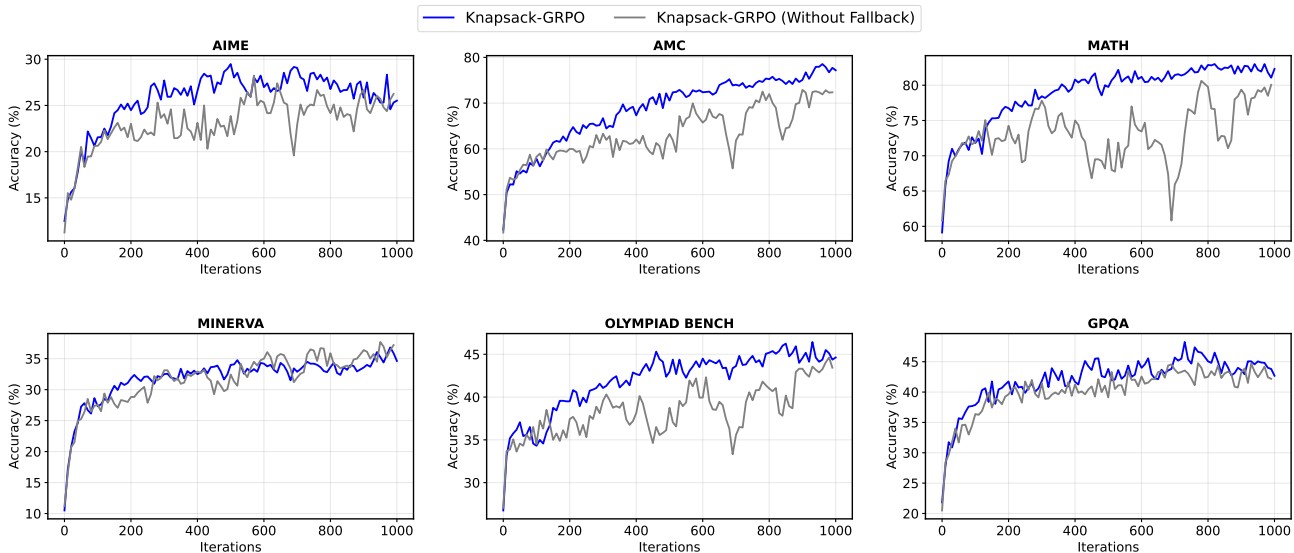

*Figure 15.* Effect of the fallback strategy. Without it, exploration budgets are disproportionately allocated to prompts with at least one successful trial, while unsolved tasks are largely ignored.

and overall degraded results. This result suggests that neglecting challenging examples during training weakens the reinforcement signal, ultimately harming the model's ability to generalize.

**Low and Up Bounds.** Our framework incorporates safeguards in the form of hyper-parameters $N_{\mathrm{low}}$ and $N_{\mathrm{up}}$, as defined in Equation (5). $N_{\mathrm{up}}$ is set to 128 primarily to facilitate faster computation of the knapsack optimization using dynamic programming; its specific value does not critically impact performance. Conversely, $N_{\mathrm{low}}$ is set to 2 to prevent degenerate allocation scenarios, particularly when success rates might be inaccurate, as elaborated in Appendix D. We present ablation results for these bounds in Figure 16, which empirically support these design choices.

**Design of Task Value.** To demonstrate the necessity of both components in our task value formulation, we conduct a controlled ablation study. We first note that removing the probability term $\mathrm{ProbNonZeroGradient}(N_i, p_i)$ would make the task value independent of the exploration budget $N_i$, rendering the optimization problem in Equation (5) underdetermined. In this degenerate case, uniform allocation trivially becomes an optimal solution, equivalent to vanilla GRPO. Therefore, we focus on examining the role of the $\mathrm{InfoGain}(p_i)$ term by comparing our full model against a variant that uses only $\mathrm{ProbNonZeroGradient}(N_i, p_i)$ as the task value.

Table 10 presents the optimal budget allocations for a synthetic task suite with uniformly distributed difficulties $p \in \{0.1, 0.2, \ldots, 0.9\}$ and a fixed total budget of $N_{\mathrm{total}} = 72$ (equivalent to 8 rollouts per task under uniform allocation). The numbers indicate how many rollouts are allocated to each difficulty level under different value function designs.

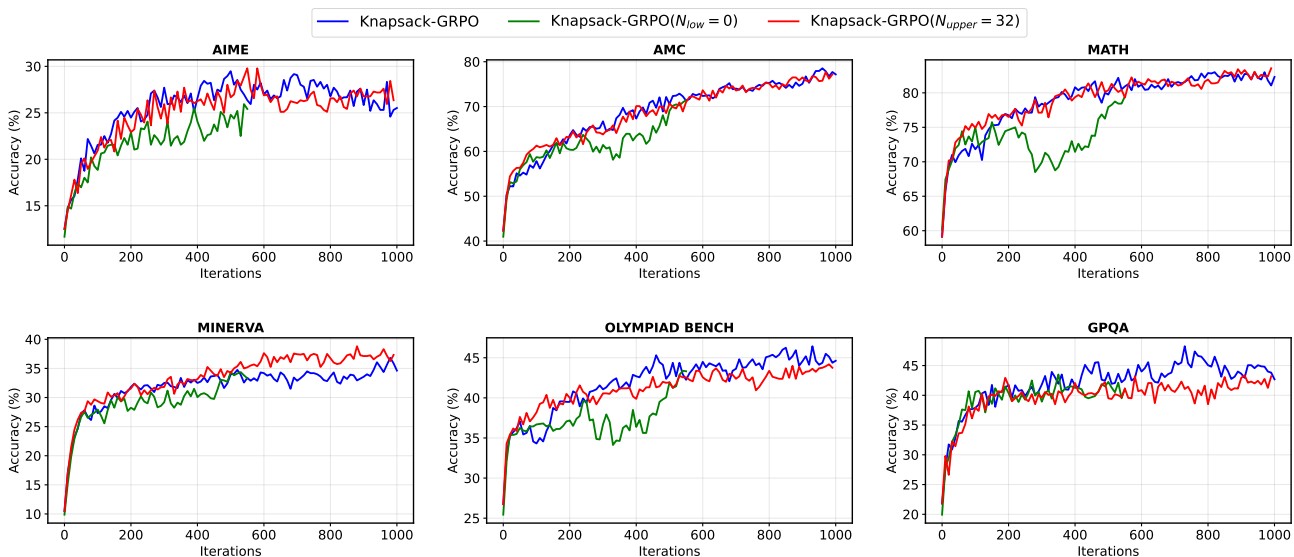

*Figure 16.* Ablation study on the impact of $N_{\mathrm{low}}$ and $N_{\mathrm{up}}$ constraints within the knapsack optimization framework.

*Table 10.* Budget allocation patterns under different value function designs. Numbers indicate rollouts allocated to tasks at each difficulty level ($p$).

| Method | Task Difficulty $p$ | | | | | | | | |
|---|---|---|---|---|---|---|---|---|---|
| | 0.1 | 0.2 | 0.3 | 0.4 | 0.5 | 0.6 | 0.7 | 0.8 | 0.9 |
| GRPO (Uniform) | 8 | 8 | 8 | 8 | 8 | 8 | 8 | 8 | 8 |
| Knapsack-GRPO | **14** | **12** | **9** | **7** | **7** | **7** | **7** | **6** | **3** |
| w/o InfoGain | 12 | 9 | 7 | 6 | 5 | 6 | 7 | 9 | 11 |

We see that using only the non-zero-gradient probability, $1 - p^N - (1-p)^N$, optimizes the diagnostic criterion from Section 3 but ignores the relative learning value of non-zero-gradient groups. This objective produces a U-shaped allocation pattern that paradoxically favors both very easy ($p \geq 0.7$) and very hard ($p \leq 0.2$) tasks. We note that over-allocating to extremely easy tasks is suboptimal because they have limited learning value. By contrast, our full model incorporates $\mathrm{InfoGain}(p_i) = p_i(1 - p_i)^2$, which concentrates budget on moderately-hard tasks ($p \in [0.1, 0.3]$) where the model struggles yet maintains sufficient success probability. This represents the optimal zone for learning—challenging enough to be informative, yet feasible enough to yield actionable gradients.

In addition, Table 11 validates these allocation patterns through downstream performance evaluation. We train Qwen2.5-Math-7B using each design variant and measure accuracy on six mathematical reasoning benchmarks. The results confirm that the $\mathrm{InfoGain}$ term is essential: removing it reduces average performance from 47.5% to 45.7% ($-1.8$ points), with consistent degradation across most benchmarks. This empirically demonstrates that effective budget allocation requires jointly considering both gradient availability (via $\mathrm{ProbNonZeroGradient}$) and learning potential (via $\mathrm{InfoGain}$).

### E.5. Interaction with Prompt Selection Baselines

We study Knapsack-GRPO alongside two prompt-selection style baselines, since prompt selection is orthogonal to rollout allocation. The first is dynamic sampling from DAPO (Yu et al., 2025), which repeatedly samples prompts and keeps those whose rollout groups contain both positive and negative rewards. The second is a PCL-style curriculum baseline inspired by Prompt Curriculum Learning (Gao et al., 2026), which uses a learned value model to identify prompts near the intermediate-difficulty regime where the model has roughly a 50% chance of success. This choice is also aligned with online difficulty filtering, which argues that balanced tasks with high success-probability variance are more informative for RLVR training (Bae et al., 2025). Our baseline does not reimplement either full system. Instead, it isolates this intermediate-difficulty preference by using the same pass-rate estimator as Knapsack-GRPO and sampling prompts from a

*Table 11.* Evaluation performance (`avg@16`) on Qwen2.5-Math-7B across value function designs. Both NonZeroProb and InfoGain terms are necessary for optimal performance.

| Method | AIME | AMC | MATH | MINERVA | OLYMPIAD | GPQA | Average |
|---|---|---|---|---|---|---|---|
| Qwen2.5-Math-7B | 12.3 | 41.0 | 61.2 | 11.8 | 26.1 | 22.0 | 26.7 |
| + GRPO | 23.9 | 70.6 | 81.7 | 33.6 | 41.9 | 40.8 | 45.2 |
| + Knapsack-GRPO | **24.3** | **77.4** | 83.9 | **34.5** | **44.1** | 43.8 | **47.5** |
| w/o InfoGain | 22.7 | 71.8 | 81.5 | 32.4 | 41.9 | 47.0 | 45.7 |

weighted distribution with weights $p(1 - p)$, a soft criterion maximized at $p = 0.5$. These methods decide *which prompts* enter training, whereas Knapsack-GRPO decides *how many rollouts* each selected prompt receives.

Table 12 reports results on Qwen2.5-Math-7B trained on DAPO-Math-17K. Both prompt-selection baselines improve over vanilla GRPO, confirming that better prompt composition helps. Knapsack-GRPO provides a larger gain in this setting, and combining it with PCL-style selection yields the best average performance, although the additional improvement over Knapsack-GRPO alone is modest. The same pattern appears in the effective-gradient ratio: PCL-style sampling increases EGR from roughly 30% to 52%, Knapsack-GRPO increases it to about 65%, and their combination reaches about 70%. This supports the interpretation that prompt selection and rollout allocation are compatible, but their empirical gains are not fully additive because both mechanisms partly target the same effective-gradient bottleneck.

*Table 12.* Interaction with prompt-selection baselines on Qwen2.5-Math-7B. All methods use the same evaluation protocol as Table 1.

| Method | AIME | AMC | MATH | MINERVA | OLYMPIAD | GPQA | Avg |
|---|---|---|---|---|---|---|---|
| GRPO | 23.9 | 70.6 | 81.7 | 33.6 | 41.9 | 40.8 | 45.2 |
| GRPO + DAPO dynamic sampling | 26.7 | 70.7 | 79.6 | 34.2 | 42.8 | 42.7 | 46.2 |
| GRPO + PCL-style | 23.5 | 74.9 | 82.6 | 35.8 | 42.1 | 39.9 | 46.1 |
| GRPO + Knapsack | 24.3 | 77.4 | 83.9 | 34.5 | 44.1 | 43.8 | 47.5 |
| GRPO + Knapsack + PCL-style | 24.7 | 76.0 | 84.4 | 36.4 | 45.2 | 42.5 | 47.8 |

For historical consistency, the Avg column averages seven task scores, while the AIME column reports the average of AIME 2024 and AIME 2025. The DAPO dynamic sampling row is read approximately from Figure 18.

Dynamic sampling operates on a different principle than our knapsack-based approach. Dynamic sampling scales the number of effective **prompts** that enter updates, while Knapsack-GRPO scales the number of effective rollout **responses** assigned within selected prompts. Since these two approaches are compatible, we conducted empirical studies to explore their interaction in more detail.

We evaluated the performance of these methods using two different metrics, as shown in the training curves in Figure 17 and Figure 18. Because dynamic sampling requires multiple exploration steps to accumulate enough effective prompts for a single gradient update, we can analyze performance in two ways:

- By exploration budget: Figure 17 shows performance relative to the total number of exploration iterations. This measures how effectively the exploration budget is converted into performance gains. We found that dynamic sampling boosts GRPO's performance on benchmarks like AIME and OLYMPIAD, improving the average score from 45.2 to 46.2. When we combined dynamic sampling with our knapsack-based exploration, performance on the AMC benchmark improved significantly (from 69.8 to 73.0), resulting in a total performance of 46.5. This is slightly better than dynamic sampling alone but worse than Knapsack-GRPO alone. We attribute this partially to the fact that Knapsack-GRPO uses more gradient iterations under the same exploration-iteration view, and therefore do not consider this a negative result.

- By gradient update iterations: Figure 18 displays performance against the number of gradient updates. This metric assesses the value of each gradient update. The results clearly show that effective gradients, whether from dynamic sampling or our knapsack-based exploration, lead to greater performance gains for the same number of update iterations, which validates the core motivation behind both techniques.

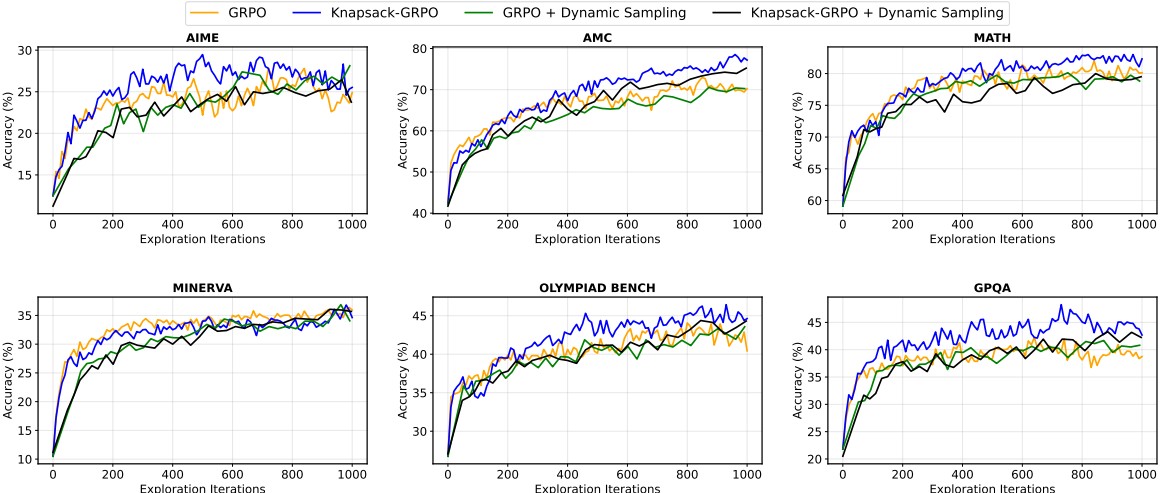

*Figure 17.* Performance of Qwen2.5-Math-7B relative to the number of exploration iterations, demonstrating how effectively the total computation budget is converted into performance gains.

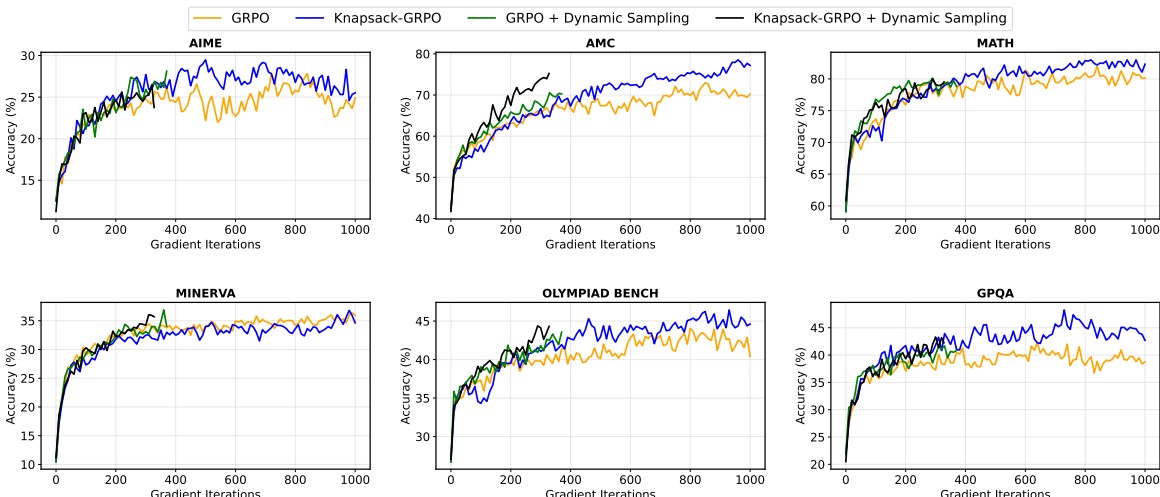

*Figure 18.* Performance of Qwen2.5-Math-7B as a function of the number of LLM gradient updates. This figure validates that effective gradients, derived from either dynamic sampling or the knapsack-based approach, lead to greater performance gains for the same number of updates.

