# OpenReview forum: "Knapsack RL: Compute-Efficient Reinforcement Learning via Heterogeneous Rollout Allocation"
_ICML.cc/2026/Conference — ICML 2026 regular_

### Official Review · Reviewer_hsV1 · 2026-03-03

**Soundness:** 3
**Presentation:** 3
**Significance:** 3
**Originality:** 3
**Overall Recommendation:** 4
**Confidence:** 3

**Summary:**

The paper first analyzes the dilemma of RL and the computational cost. They then propose Knapsack RL to dynamically assign the number of rollouts to each prompt according to its potential value.

**Compliance With Llm Reviewing Policy:**

Affirmed.

**Key Questions For Authors:**

See above

**Limitations:**

yes

**Strengths And Weaknesses:**

### **Strength**
- The authors convert the budget allocation problem into a constrained optimization and provide a theoretical explanation.
- The authors provide detailed experiments on various models to show that the method is robust.

### **Weakness**
- Could the authors provide the rollout time of each step of KnapSack RL compared with standard GRPO? Does this method incur more severe long-tail generation?
- The model size is limited to 7 B. Could the authors carry out experiments on larger models, like Qwen2.5-14B?

---

> ### Author Rebuttal · Authors · 2026-03-31
>
> We thank Reviewer hsV1 for the positive review and practical questions!
>
> **W1: Rollout time comparison between Knapsack RL and standard GRPO. Does this method incur more severe long-tail generation?**
>
> Thank you for this practical question. We measured the wall-clock time on both a short-CoT model (Qwen2.5-Math-7B) and a long-CoT model (DeepSeek-R1-Distill-1.5B):
>
> | Method                           | Avg Step Time (s) | Rollout Time (s) |
> | -------------------------------- | ----------------: | ----------------: |
> | Qwen2.5-7B + GRPO          |               110 |                43 |
> | Qwen2.5-7B + Knapsack GRPO |               115 |                40 |
> | DPSK-R1-1.5B + GRPO             |               239 |               112 |
> | DPSK-R1-1.5B + Knapsack GRPO    |          249 |               120 |
>
> For the short-CoT model, where response lengths are roughly comparable across easy and hard tasks, the overhead is negligible. For the long-CoT model, the increase in step time is less than 4%.
>
> This modest overhead can be explained as follows. The maximum rollout time within a batch is determined by the *longest* response, which both methods face equally. Furthermore, allocating more rollouts to certain prompts does not linearly increase generation time, because auto-regressive decoding is largely memory-bandwidth-bound (a similar observation has been reported in [1]).
>
> In summary, the runtime overhead of Knapsack allocation is minimal and well within practical tolerance.
>
> **W2: Model size limited to 7B. Can the authors experiment on larger models like Qwen2.5-14B?**
>
> Yes. We have conducted experiments on Qwen2.5-14B-Instruct trained on DAPO-MATH-17K:
>
> |                 | AIME     | AMC      | MATH     | MINERVA  | OLYMPIAD | GPQA | Avg      |
> | --------------- | -------- | -------- | -------- | -------- | -------- | ---- | -------- |
> | Qwen2.5-14B     | 12.0     | 50.5     | 75.0     | 32.8     | 41.4     | 40.9 | 42.1     |
> | + GRPO          | 14.8     | 59.1     | 76.4     | 35.2     | **46.8**     | 43.0 | 45.9     |
> | + Knapsack GRPO | **16.4** | **62.0** | **78.8** | **36.4** | 45.9     | 44.8 | **47.4** |
>
> Knapsack GRPO consistently outperforms standard GRPO on 5 out of 6 benchmarks, with gains of +1.6 on AIME, +2.4 on MATH, and +2.9 on AMC. This demonstrates that our method scales effectively to larger models. In addition, we also observe that the effective gradient ratio improves from 34.0% to 62.2% with Knapsack GRPO, a signal indicating its generalization to larger models.
>
> We sincerely hope the above responses address your concerns. Should you have any further questions, we would be happy to discuss them.
>
> ---
>
> [1] Liu, Mingjie, et al. "Prorl: Prolonged reinforcement learning expands reasoning boundaries in large language models." *arXiv preprint arXiv:2505.24864* (2025).

---

> > ### Author Rebuttal · Reviewer_hsV1 · 2026-04-03
> >
> > Thank the authors for their rebuttal. I will remain my original positive score.

---

> > > ### Author Response · Authors · 2026-04-03
> > >
> > > Dear Reviewer hsV1,
> > >
> > > We are pleased to see that your concerns have been addressed! Thank you very much for taking the time to review our paper. We will update the manuscript accordingly and sincerely appreciate your positive and support for our work.

---

### Official Review · Reviewer_CK6R · 2026-03-09

**Soundness:** 3
**Presentation:** 3
**Significance:** 2
**Originality:** 3
**Overall Recommendation:** 4
**Confidence:** 4

**Summary:**

The paper studies critic-free policy optimization algorithms applied to training LLM reasoning models. Usually, those algorithms allocate fixed and equal amount of budget per problem instance to calculate the gradient update, e.g., N rollouts per instance to estimate advantages in GRPO. Authors argue that this fixed budget allocation leads is inefficient at extracting learning signal. Consequently, instead of having fixed number of rollouts per problem instance, authors propose to allocate an adaptive number of rollouts to each instance maximizing learning progress. Practically, authors frame this allocation problem as knapsack problem, derive the value function based on the expected improvement that both reasonably assesses the learning progress and allows for efficient calculation of the optimal allocation. This results in the module that can be incorporated into existing algorithms with minimal changes. Authors instantiate their approach based on the GRPO algorithm and show that it leads to better results on standard LLM reasoning benchmarks under the same budget when compared to vanilla GRPO that assigns equal budget to each problem instance.

**Compliance With Llm Reviewing Policy:**

Affirmed.

**Final Justification:**

I thank the authors for the detailed rebuttal. I believe this paper provides novel view on constructing reinforcement learning algorithms for LLM reasoning and performs broad evaluation, thus, it is well-suited for the acceptance.

**Key Questions For Authors:**

See weaknesses.

**Limitations:**

yes

**Strengths And Weaknesses:**

### Strengths

1. The problem formulation is very compelling. Authors do very good work at guiding the reader through the experimental evidence of the problem in Section 3 and suggest a well-motivated framework to tackle this problem in Section 4.
2. Experimental coverage is broad. Authors study the effectiveness of their approach with models ranging from 1.5B to 7B and of different families. Furthermore, ablations of different parts of the methodology are included, providing additional support.

### Weaknesses

1. The paper explicitly notes that rollout lengths vary across tasks, yet the knapsack formulation treats every rollout as having equal cost. In practice, hard tasks often generate longer reasoning chains, so the actual compute cost of allocating e.g. 93 rollouts to a hard prompt may be substantially more than 93/8 times the cost of a standard allocation. This can undermine the "free lunch" that the proposed methodology claims to some degree. It would be nice to see the average token counts for high-budget vs. low-budget prompts to quantify this effect.
2. The InfoGain approximation (Appendix B.2) relies on too much of unrealistic assumptions, e.g., unit learning rate, unit advantage, softmax parametrization over actions. Real LLM generation and training is much more sophisticated, and deviates from those assumptions. I understand the urge to provide theoretical justification, but it shouldnt appear at the cost of too much simplifications. The paper would greatly benefit from providing the experimental evidence of approximation quality under the real setting.
3. Currently, InfoGain approximation + the overall value function design feels a little bit "reverse engineered" to achieve good properties such as monotonicity and diminishing returns to allow for the greedy solver for the knapsack problem. It would be nice to discuss the ideal value formulation, describe why the ideal case may be problematic (e.g. knapsack problem solver unfeasibility / inefficiency) and then proceed with the proposed solution. Maybe even provide some experiment showing that the proposed solution would not lose much to the ideal case.

---

> ### Author Rebuttal · Authors · 2026-03-31
>
> We thank Reviewer CK6R for the positive evaluation and suggestions!
>
> **W1: Rollout lengths vary across tasks, but the knapsack formulation treats every rollout as equal cost. Hard tasks may generate longer reasoning chains, undermining the "free lunch" claim.**
>
> Thanks for this careful observation. Our method usually assigns more rollout budget to medium- and hard-difficulty prompts, which often require more tokens (especially for long-CoT models). Thus, from a computational perspective, our "free lunch" claim is imprecise; it applies more at the algorithmic level. We computed the average token counts for prompts receiving high vs. low rollout budgets:
>
> |              | Avg Tokens of High-Budget Prompts | Avg Tokens of Low-Budget Prompts |
> | ------------ | --------------------------------: | -------------------------------: |
> | Qwen2.5-7B   |                             1,135 |                              953 |
> | DPSK-R1-1.5B |                             6,614 |                            4,556 |
>
> However, allocating more rollouts to medium- and hard-difficulty prompts does not incur substantial overhead (less than 4%; see our response to Reviewer hsV1). This is partly because: (1) both uniform and adaptive allocation face the same long-tail latency bottleneck, with step time governed by the longest response in a batch; and (2) auto-regressive generation is memory-bandwidth-bound, so extra rollouts do not linearly increase wall-clock time.
>
> We will revise our claim as follows: *Knapsack allocation allows larger budgets for informative prompts. In practice, this comes at a modest cost when those prompts also require more tokens.*
>
> **W2: InfoGain approximation relies on unrealistic assumptions (unit learning rate, unit advantage, softmax parametrization).**
>
> We understand this gap between theory and practice:
>
> 1. *Role of the approximation.* The InfoGain approximation serves as a *design principle* rather than a precise quantitative prediction. Its purpose is to identify the qualitative structure that moderately difficult prompts yield the highest marginal learning signal, guiding the value function design.
>
> 2. *Empirical validation.* Despite the simplifying assumptions, the predicted allocation pattern (more rollouts to moderate-difficulty prompts, fewer to very easy/hard ones) aligns well with our empirical results (see Figure 6 and additional results in Appendix E).
>
> 3. *Simulation evidence.* We conducted simulations to assess the approximation quality. Specifically, we fine-tune Qwen2.5-Math-7B with a single prompt and positive examples only (consistent with InfoGain), then measure the accuracy improvement on the same prompt. We group prompts by success rate bins and report the average improvement:
>
>    |                        | 0.1  | 0.2  | 0.3  | 0.4  | 0.5  | 0.6  | 0.7  | 0.8  | 0.9  |
>    | ---------------------- | ---- | ---- | ---- | ---- | ---- | ---- | ---- | ---- | ---- |
>    | Empirical measurement  | 0.06 | 0.13 | 0.22 | 0.10 | 0.14 | 0.06 | 0.05 | 0.07 | 0.03 |
>    | Theoretical prediction | 0.08 | 0.13 | 0.15 | 0.14 | 0.13 | 0.10 | 0.06 | 0.03 | 0.01 |
>
>    The empirical trend is broadly consistent with our theoretical prediction: the largest improvement is concentrated in $p \in [0.2, 0.5]$, while gains diminish as $p$ approaches 0 or 1. This validates the qualitative correctness of our approximation despite the simplifying assumptions.
>
> **W3: The value function design feels "reverse engineered" to achieve monotonicity and diminishing returns for the greedy solver. Suggest discussing the ideal formulation and why it is problematic.**
>
> This is a constructive suggestion. We will restructure the presentation as follows:
>
> 1. *Ideal value formulation.* The ideal value of allocating $N$ rollouts to a prompt would be the *expected policy improvement* after a gradient update using those $N$ rollouts. Computing this exactly requires: (a) generating $N$ rollouts, (b) computing the gradient, (c) performing a parameter update, and (d) evaluating the updated policy. This is computationally prohibitive.
>
> 2. *Why the ideal formulation is problematic for knapsack solving.* Even if we could compute the ideal value, it would not necessarily satisfy diminishing returns (submodularity), which the greedy knapsack solver relies on for near-optimality. Without this property, the problem may become NP-hard and require exponential-time solvers.
>
> 3. *Our approximation as a principled middle ground.* Our first-order Taylor expansion naturally yields a value function with the desired structural properties (monotonicity, diminishing returns) because they are inherent to the information-theoretic structure: each additional rollout provides diminishing marginal information about the prompt's reward distribution. In our view, this is not "reverse engineering" but a consequence of the chosen abstraction.
>
> ---
>
> We will revise the paper accordingly. We hope these responses could address your concerns and welcome suggestions.

---

> > ### Author Rebuttal · Reviewer_CK6R · 2026-04-02
> >
> > I thank the authors for the provided response and for the adjustment of claims. I keep my original score of leaning towards accepting this paper.

---

> > > ### Author Response · Authors · 2026-04-03
> > >
> > > Dear Reviewer CK6R,
> > >
> > > Thank you very much for your follow-up and for maintaining your recommendation toward accepting our paper! We greatly appreciate your thoughtful feedback and the time you have spent evaluating our work. Your support gives us a valuable opportunity to share our results with the community and contribute to further progress in this area.
> > >
> > > From your response, we were encouraged to see that our rebuttal and clarification of our claims *appear to have addressed much of your concern*. At the same time, we noticed that you selected the option indicating that the concerns are “partially resolved or unresolved, but the remaining concerns are not easily addressed in a short rebuttal.” **We wanted to respectfully ask whether this reflects any remaining substantive concerns about the paper, or whether it may have been selected inadvertently.**
> > >
> > > If there are still important issues that you feel have not been fully addressed, we would be very grateful if you could briefly indicate which concerns remain open and, if possible, suggest how we might further improve the paper in revision. Your guidance would be extremely valuable in helping us strengthen the final version of the work. We understand that, under ICML policy, we are unable to respond further at this stage. If possible, **we would greatly appreciate any additional comments, either in a direct reply or through an update to your review.**
> > >
> > > Thank you again for your consideration and support.

---

### Official Review · Reviewer_jNkD · 2026-03-11

**Soundness:** 2
**Presentation:** 3
**Significance:** 3
**Originality:** 2
**Overall Recommendation:** 3
**Confidence:** 4

**Summary:**

This work studies the rollout allocation problem in LLM reinforcement finetuning. The authors frame it as a knapsack problem, aiming to maximize a self-defined value function by dynamically allocating rollouts for different prompts, with the total number of rollouts as a constraint. Empirical results show that the method enhances the effective gradient ratio and accuracy compared to GRPO.

**Compliance With Llm Reviewing Policy:**

Affirmed.

**Final Justification:**

My concern "W1: Missing end-to-end comparisons with efficient prompt selection methods" remains unaddressed. The additional results appear to be from a self-defined toy setting and lack sufficient implementation details. I find it difficult to understand how these experiments were implemented. Since there are many existing methods aiming for the same goal (efficient rollout and enhancing effective gradient ratio), I find it necessary to include a direct comparison. However, the authors fail to provide a comparison against any of these related methods on any backbone. Therefore, I lean toward recommending rejection.

**Key Questions For Authors:**

1. Can the authors add a comparison with other efficient rollout methods [1-6] (at least several lastest of them) in Table 1 and Figure 6 to show the advantage of the proposed method?

2. Can the authors conduct more experiments other than math datasets to demonstrate the generalizability of the proposed method?

3. Can the authors prove that the definition of the Value function used in the paper is optimal?

4. In Figure 8, the authors show the biggest InfoGain is achieved when the success rate is around 0.3-0.4, which contradicts most previous works showing 0.5 to be the most beneficial. Can the authors explain the reason for this discrepancy? Which one is better?

**Limitations:**

See Weaknesses and Questions.

**Strengths And Weaknesses:**

**Strengths:**

1. The studied problem of rollout allocation is crucial for efficient LLM reinforcement finetuning.

2. The idea of dynamically allocating rollouts for different prompt and framing the task as a knapsack problem is reasonable and interesting.

3. The paper is well-written and easy to follow.

**Weaknesses:**

Major:

1. The evaluation is not comprehensive. There are many existing works aiming for efficient rollouts, including but not limited to [1-6], but none of them are included for comparison. The sole comparison with vanilla GRPO is insufficient, making the effectiveness of the proposed method unclear. Furthermore, the model is only trained on math reasoning datasets, making it unclear whether the proposed method can generalize to other reasoning scenarios such as code and logic.

2. The method requires an initial epoch of homogeneous allocation to estimate $p_i$, which narrows the applicable scenarios. If only one or a few epochs are needed for training, the method cannot achieve effective acceleration.

3. The effectiveness of the method highly relies on the definition of the Value function, but the derivations are based on heuristic assumptions. There could be many reasonable definitions, but the authors do not show that the choice in the paper is optimal.

Minor:

4. There are many reference errors in the Appendix (displaying as ??); the authors should carefully proofread the manuscript before submission.

[1] Improving data efficiency for llm reinforcement fine-tuning through difficulty-targeted online data selection and rollout replay

[2] Act only when it pays: Efficient reinforcement learning for llm reasoning via selective
rollouts

[3] Can prompt difficulty be online predicted for accelerating rl finetuning of reasoning models?

[4] Prompt Curriculum Learning for Efficient LLM Post-Training

[5] BOTS: A Unified Framework for Bayesian Online Task Selection in LLM Reinforcement Finetuning

[6] Dynamics-Predictive Sampling for Active RL Finetuning of Large Reasoning Models

---

> ### Author Rebuttal · Authors · 2026-03-31
>
> We thank Reviewer jNkD for the helpful review and references!
>
> **W1: Missing comparisons with [1-6].**
>
> Thank you for bringing these works to our attention. We appreciate [1-6] for showing that prompt selection and curriculum design also matter for effective gradients, and will cite and discuss them in the revision. However, [1-6] focus on **prompt/data selection** (*which* prompts to train on), while our method focuses on **budget allocation** (*how many* rollouts per prompt). These are **complementary**: one can first select prompts, then allocate rollouts optimally. We validated this with DAPO's dynamic sampling (Appendix E.5) and with curriculum-based selection from [1, 4]:
>
> |                                | AIME | AMC  | MATH | MINERVA | OLYMPIAD | GPQA | Avg  |
> | ------------------------------ | ---- | ---- | ---- | ------- | -------- | ---- | ---- |
> | + GRPO                         | 23.9 | 70.6 | 81.7 | 33.6    | 41.9     | 40.8 | 45.2 |
> | + GRPO + Curriculum            | 23.5 | 74.9 | 82.6 | 35.8    | 42.1     | 39.9 | 46.1 |
> | + GRPO + Knapsack              | 24.3 | 77.4 | 83.9 | 34.5    | 44.1     | 43.8 | 47.5 |
> | + GRPO + Knapsack + Curriculum | 24.7 | 76.0 | 84.4 | 36.4    | 45.2     | 42.5 | 47.8 |
>
> Combining both gives the best result (47.8), confirming orthogonality.
>
> **W2: Only math reasoning; unclear generalization.**
>
> We conducted new experiments on code and logic tasks.
>
> *Code generation* (Qwen2.5-7B on [PRIME-RL] code split):
>
> |                 | Taco | Codeforces | Codecontests | Apps | Livecodebench | Avg  |
> | --------------- | ---- | ---------- | ------------ | ---- | ------------- | ---- |
> | + GRPO          | 18.0 | **21.3**   | 31.0         | 32.8 | **22.8**      | 24.8 |
> | + Knapsack GRPO | **23.0** | 20.3  | **34.0**     | **36.0** | 22.3     | **26.3** |
>
> Knapsack-GRPO outperforms GRPO on 3/5 benchmarks (+5 Taco, +3 Codecontests, +3.2 Apps).
>
> *Logic tasks* (Qwen2.5-7B on [SynLogic] easy split, averaged over 27 tasks):
>
> |                 | Arc_agi | calcudoko | Math_path | Sudoku | Web_of_lies | Avg  |
> | --------------- | ------- | --------- | --------- | ------ | ----------- | ---- |
> | + GRPO          | 8.3     | 11.3      | 78.9      | 72.5   | 15.0        | 47.6 |
> | + Knapsack GRPO | **22.2** | **15.0** | **86.3** | **85.0** | 12.5     | **52.5** |
>
> Knapsack-GRPO achieves +4.9 avg improvement, with the effective gradient ratio increasing ~50% on logic tasks confirming generalization across RLVR domains.
>
> **W3: Requires initial epoch to estimate $p$; limits applicability.**
>
> Valid concern. Three practical solutions exist: (1)  *Pre-existing estimates*: industry pipelines often evaluate model performance on training prompts during data curation; these estimates can initialize the Knapsack solver without warm-up. (2) *Lightweight warm-start*: only N=4 samples per prompt provides sufficient signal, as in [PCL]. (3) *Online prediction*: logistic regression on prompt embeddings can predict $p$ from the first batch (Appendix E.3), removing the cold-start entirely. We will discuss these in the revision.
>
> **W4: Value function is heuristic; not shown optimal.**
>
> Our InfoGain derivation is a principled first-order Taylor expansion of expected policy improvement. While assumptions (unit learning rate, softmax parametrization) are simplifying, they capture the essential structure. Computing the "ideal" information gain would require backward passes per prompt plus extra rollouts to estimate improvement, making it prohibitive in practice. The monotonicity and diminishing-returns properties are natural consequences of the information-theoretic structure, enabling an efficient greedy solver. Empirically, our ablation study (Appendix E.4) shows that both components (ProbNonZeroGradient and InfoGain) are essential. Results across models (1.5B–8B), families (Qwen, Llama), and benchmarks further validate the design.
>
> **W5: Reference errors.**
>
> Apologies for this oversight. We will fix all LaTeX errors in the revision.
>
> **Q6: InfoGain peaks at 0.3–0.4 vs. 0.5 in prior work.**
>
> Thank you for this careful observation. The difference stems from the modeling assumptions. Prior works often relate the expected learning benefit to reward variance or signal-to-noise ratio, which peaks at $p=0.5$. In contrast, our formulation decomposes the expectation into zero-advantage and non-zero-advantage events. Conditioning on the non-zero-advantage case, and accounting for the softmax geometry, a success at $p=0.3$–$0.4$ is more informative than one at $p=0.5$. This leads to a mild bias toward harder tasks, which is consistent with [GVM].
>
> We hope these responses address your concerns and welcome further questions.
>
> ---
>
> [PRIME-RL] Cui et al., arXiv:2502.01456 (2025).
> [SynLogic] Liu et al., arXiv:2505.19641 (2025).
> [PCL] Gao et al., arXiv: 2510.01135 (2025).
> [GVM] Yao et al., arXiv:2505.02391 (2025).

---

> > ### Author Rebuttal · Reviewer_jNkD · 2026-04-01
> >
> > I thank the authors for the response, which has partially addressed "W2: Only math reasoning; unclear generalization" and "W3: Requires initial epoch to estimate p; limits applicability." Although neither is fully resolved (W2 would benefit from additional baselines for comparison, and the response to W3 discusses mitigation rather than a complete solution), the current form may be acceptable.
> >
> > My primary remaining concern is "W1: Missing comparisons with [1-6]." In the reported results, the performance gap between "+ GRPO + Knapsack (47.5)" and "+ GRPO + Knapsack + Curriculum (47.8)" is marginal and could plausibly be attributed to random seed variation alone. This raises doubt about the claimed complementarity between the proposed prompt/data selection and budget allocation methods. Moreover, both categories of methods share the same underlying goal of enhancing the effective gradient ratio, which makes a direct comparison all the more necessary. I would expect to see an end-to-end comparison with several latest methods in [1-6], with updated results incorporated into Table 1 and Figure 6 for at least one backbone.
> >
> > Additionally, the literature review appears insufficient, as many relevant efficient rollout methods are not cited (include but not limited to [1-6]). I leave it to the authors to address this in revision.

---

> > > ### Author Response · Authors · 2026-04-03
> > >
> > > Dear Reviewer jNkD,
> > >
> > > Thank you very much for your thoughtful follow-up! We are encouraged that our previous rebuttal has partially addressed your concerns about **W2** and **W3**, and we are happy to clarify the remaining issues.
> > >
> > > **Remaining concern on W1:** To make the discussion clearer, we separate the follow-up into two sub-concerns.
> > >
> > > > **W1(a): Missing end-to-end comparisons with efficient prompt selection methods**
> > >
> > > **Response to W1(a):** We appreciate this suggestion. However, methods in [1–6] are **closely related** to our work, but they are **not fully aligned baselines** for the main question studied here. Our method focuses on **heterogeneous rollout budget allocation under a fixed global compute budget**, whereas many recent efficient RL finetuning methods focus more on **prompt/task selection**, **difficulty prediction**, or **pre-rollout filtering**.
> > >
> > > That said, we agree there should be a clearer empirical connection to this line of work, and **we have tried to provide such a bridge in two ways**.
> > >
> > > - First, in **Appendix E**, we already compared against **DAPO's dynamic sampling** (the first work in this line). As discussed there, dynamic sampling and our method operate on different parts of the pipeline and can also be combined.
> > >
> > > - Second, in the **previous response**, we also implemented a **curriculum-style baseline** following [1] and [4], namely prioritizing prompts using the reward variance \(p(1-p)\). On Qwen2.5-Math-7B / DAPO-MATH-17K, we observed:
> > >
> > >   - **+GRPO:** 45.2
> > >   - **+GRPO + Curriculum:** 46.1
> > >   - **+GRPO + Knapsack:** 47.5
> > >
> > > These results suggest that curriculum-style selection is helpful, but knapsack-based rollout allocation yields a larger gain in our setting. The same pattern is also reflected in the effective gradient ratio: curriculum improves it from **30% to 52%**, while knapsack improves it further to **65%**. We hope this addresses the reviewer’s question directly.
> > >
> > > At the same time, we would like to note that **a fully fair end-to-end sweep over all methods in [1–6] is nontrivial during the rebuttal window**, because these methods introduce components such as replay, difficulty predictors, and different Verl/vLLM versions, all of which require nontrivial integration into training infrastructure. **Some work in this line is also extremely recent**; for example, **[6] (arXiv 2603.10887) appeared after ICML submission**, so it could not have been included in the original comparison. Therefore, while we agree such comparisons are valuable, we are not able to complete all of them reliably within the rebuttal period, and plan to add more results in the revision.
> > >
> > > > **W1(b): Are knapsack-based allocation and prompt selection methods truly complementary?**
> > >
> > > **Response to W1(b):** We appreciate this follow-up and agree that our previous wording on “complementarity” should be made more precise.
> > >
> > > - Conceptually, the two approaches operate on different parts of the RL pipeline. Prompt selection methods modify the prompt sampling distribution $\rho(x)$, i.e., **which prompts are selected for training**, whereas knapsack-based allocation determines the rollout budget $N_i$ for each selected prompt, i.e., **how much compute is spent on each selected prompt**. In this sense, the two mechanisms are orthogonal at the level of formulation.
> > >
> > > - Empirically, however, we agree that their practical complementarity is **limited rather than substantial** in our current setting. The combined result (**+GRPO + Knapsack + Curriculum = 47.8**) is only slightly better than **+GRPO + Knapsack = 47.5**. We do not believe this small gap is merely random noise. Rather, it is consistent with the intermediate statistics: curriculum improves the effective gradient ratio to about **52%**, knapsack to about **65%**, and their combination to about **70%**. This suggests that the two methods remain compatible, but that the additional benefit of curriculum on top of knapsack is modest, which in turn explains the limited final performance gain.
> > >
> > > **Response to related work:** Thank you for emphasizing this point. We agree that the related-work discussion should be improved.
> > >
> > > In the current draft, Section 6 and Appendix A already discuss prompt selection and curriculum learning. In revision, we will expand this discussion to better cover **[1–6]** and clarify the distinction between:
> > >
> > > - methods that improve efficiency mainly through **prompt/task selection**, **difficulty prediction**, **pre-rollout filtering**, or **rollout replay**, and
> > > - our method, which focuses on **heterogeneous rollout budget allocation under a fixed global compute budget**.
> > >
> > > Moreover, if the reviewer feels there are important works beyond [1–6] that should be discussed, please let us know in your updated review (since we cannot further repond according to ICML policy).
> > >
> > > ---
> > >
> > > Thank you again for your thoughtful follow-up. We sincerely appreciate your time and consideration.

---

### Official Review · Reviewer_PiU4 · 2026-03-13

**Soundness:** 3
**Presentation:** 4
**Significance:** 3
**Originality:** 3
**Overall Recommendation:** 4
**Confidence:** 4

**Summary:**

The work focuses on the concept of optimizing the exploration budget allocation across different tasks (prompts) during the Reinforcement Learning (RL) fine-tuning of Large Language Models (LLMs). The authors observe that uniform rollout allocation (e.g., $N=8$ for all prompts) is highly inefficient for policy gradient algorithms like GRPO, as it yields zero-gradient updates for both trivially easy tasks (where all rollouts are correct) and hopelessly hard tasks (where all rollouts are incorrect).  To address this, the paper formulates the prompt-level rollout allocation as a Knapsack optimization problem. By defining the "learning value" of a prompt as the probability of a non-zero gradient multiplied by the expected information gain (derived via a Taylor expansion approximation), and treating each rollout as a unit cost, the method dynamically allocates larger budgets to moderately challenging tasks. Empirical results on mathematical reasoning benchmarks using various models (e.g., Qwen, Llama) demonstrate that this Knapsack-GRPO method improves sample efficiency, increases the effective gradient ratio, and yields higher asymptotic performance compared to standard GRPO.

**Compliance With Llm Reviewing Policy:**

Affirmed.

**Final Justification:**

I have seen the author’s response and efforts, but I am inclined to keep my score unchanged.

**Key Questions For Authors:**

How does the Knapsack formulation hold up if we apply it to continuous scalar rewards (e.g., standard RLHF with a reward model) rather than rule-based binary rewards? Specifically, wouldn't the Taylor expansion approximation for InfoGain ($p(1-p)^2$) derived in Appendix B.2 completely break down in this continuous setting? How would you adapt the ProbNonZeroGradient and InfoGain components?

**Limitations:**

Yes

**Strengths And Weaknesses:**

**Strengths:**

1. While much recent attention has been on scaling test-time compute, optimizing training-time exploration is just as critical. This paper tackles a major computational bottleneck in modern LLM reasoning post-training and provides a highly practical solution.
2. Mapping the rollout budget problem to a classic discrete Knapsack problem is a creative and elegant application of operations research. The proposed algorithm is computationally lightweight and plugs easily into existing frameworks.
3. The experiments are comprehensive, covering multiple reasoning benchmarks (AIME, AMC, MATH, etc.) and model scales (1.5B to 8B). The absolute performance gains (e.g., +2-4% on average) are non-trivial and robustly demonstrated.

**Weaknesses:**

1. The core method relies on empirical success rates ($p_i$) calculated from the last epoch.  However, LLM policies can shift rapidly during RL. Using stale $p_i$ estimates introduces a distribution shift that might misguide the Knapsack solver. While the authors did explore an online generation baseline and a preliminary logistic regression predictor in Appendix E.3, these are not fully integrated into the main pipeline.  Relying on delayed signals limits the method's theoretical soundness in practice.
2. The paper is entirely evaluated on math/reasoning tasks with sparse binary rewards (1 for correct, 0 for incorrect).  It is unclear if or how this framework generalizes to standard RLHF (e.g., helpfulness/harmlessness alignment) where rewards are continuous scalars outputted by a reward model.

---

> ### Author Rebuttal · Authors · 2026-03-31
>
> We thank Reviewer PiU4 for the positive assessment and thoughtful questions!
>
> **W1: Reliance on a delayed estimator for success rate $p$.**
>
> Thank you for this important observation. We acknowledge that there are many design choices in estimating the success rate $p$ and in practice it often involves a trade-off between accuracy and computational cost. As shown in our paper, the success rate distribution shifts gradually across consecutive epochs (see Appendix E), meaning the previous epoch's estimates remain a reasonable proxy.  In addition to our design, we would like to highlight several practical mitigations to further reduce this distribution shift:
>
> 1. *Warm-start estimation.* Before entering the Knapsack allocation phase, one can perform a lightweight warm-start by rolling out a small number of samples (e.g., N=4) per prompt to obtain a more up-to-date estimate of $p$. A similar warm-start strategy has been adopted in prior work [1]. The remaining budget is then allocated via the knapsack solver.
> 2. *Online estimation (future work).* We agree that integrating online logistic regression predictors (as explored in Appendix E.3) into the main pipeline is a promising direction. This would allow continuous refinement of $p$ estimates during training. However, this requires additional training-infrastructure engineering, so we plan to investigate it more thoroughly in the future revision.
>
> In summary, we believe our current design strikes a reasonable balance between effectiveness and computational cost, while leaving room for future improvements. We will revise the paper to make this trade-off more explicit.
>
> **W2: Extension to RLHF tasks with continuous rewards.**
>
> This is an excellent question about the generalizability of our framework. We discuss two aspects:
>
> 1. *Practical relevance.* Compared with RLVR, which aims to substantially improve reasoning capabilities, standard RLHF for style/format alignment is typically lightweight—often requiring only a single epoch of training with fewer rollouts. In such settings, the uniform allocation baseline is already reasonable, and the marginal benefit of adaptive allocation may be limited.
>
> 2. *Methodological extension.* For continuous-reward RLHF, the key challenge is redefining the "information value" of rollouts, since there is no binary success/failure signal. We believe reward variance is a natural proxy: prompts with high reward variance indicate that the model's policy is uncertain in that region, suggesting high local improvement potential. Specifically, one could replace `Value(N_i, p_i)` with a variance-based measure (e.g., the variance of advantages across rollouts) and retain the knapsack formulation.
>
> **Key Question: How does the Knapsack formulation hold up with continuous scalar rewards?**
>
> The key challenge is reformulating the task value function. In the RLVR setting, we benefit from a clean decomposition into zero-advantage and non-zero-advantage events, which enables a simplified first-order analysis of policy gradients. While this decomposition does not directly carry over to continuous scalar rewards, the underlying principle remains applicable.
>
> To sketch a possible extension: consider the standard policy gradient estimator $\hat{g} = \frac{1}{N}\sum_{i=1}^{N} A_i \nabla \log \pi(y_i|x)$, where $A_i$ denotes the advantage. The expected squared norm of the gradient scales as $\|\mathbb{E}[\hat{g}]\|^2 + \frac{1}{N}\text{Var}(\hat{g})$. The first term captures the learning signal (expected improvement direction), while the second term represents estimation noise that decreases with more rollouts. For continuous rewards, one can connect InfoGain with the signal-to-noise ratio of gradients, which is naturally related to the advantage variance $\text{Var}(A_i)$ of a prompt. Prompts with high advantage variance benefit most from additional rollouts (to reduce noise), while prompts with near-zero variance (trivially easy or consistently scored) gain little. This provides a principled replacement for the binary-reward InfoGain and preserves the diminishing-returns structure needed by the knapsack solver. We believe this direction deserves further investigation but currently is beyond the main scope of our paper.
>
> [1] Gao, Zhaolin, et al. "Prompt curriculum learning for efficient llm post-training." *arXiv preprint arXiv:2510.01135* (2025).
>
> ---
>
> We sincerely hope the above responses address your concerns. Should you have any further questions, we would be happy to discuss them.

---

> > ### Author Rebuttal · Reviewer_PiU4 · 2026-04-04
> >
> > Thanks you for the rebuttal and clarifications. I still keep my score

---

> > > ### Author Response · Authors · 2026-04-04
> > >
> > > Dear Reviewer PiU4,
> > >
> > > Thank you for your follow-up and for taking the time to review our paper and rebuttal!
> > >
> > > We noticed that you selected “(b) Partially resolved,” but we were not able to identify any remaining follow-up questions in your acknowledgement. **If there are still critical concerns that we have not fully addressed, we would be very grateful if you could indicate them in your updated review.** Accoding to ICML policy, we are no longer able to respond further at this stage, but such clarification in your updated review comments would help us better address these points in a future revision.
> > >
> > > Otherwise, if you feel that our rebuttal has addressed your main concerns, we would sincerely appreciate it if you would consider updating the acknowledgement to “(a) Fully resolved.”

---

### Decision · Program_Chairs · 2026-04-30

**Decision:**

Accept (regular)

**Comment:**

This paper formulates rollout budget allocation during RL post-training as a knapsack problem, allocating more rollouts to prompts where the model can learn most effectively. The formulation is clean and the empirical improvements on effective gradient ratio are encouraging. However, reviewer jNkD raised a legitimate concern about missing comparisons with prompt selection methods that target the same underlying metric. The authors argue that prompt selection and budget allocation are distinct problems, and I find this distinction conceptually valid, though the rebuttal results showing marginal complementarity weaken the orthogonality claim somewhat. The theoretical framing via InfoGain provides useful intuition even if the underlying assumptions are acknowledged to be approximate. the authors must include direct comparisons with at least two established prompt selection methods, qualify the efficiency claims to account for longer sequences from harder prompts, and discuss the relationship between budget allocation and prompt selection more carefully.